# The Cloud Resolving Model Radar Simulator (CR-SIM) Version 3.3: Description and Applications of a Virtual Observatory

Mariko Oue[1], Aleksandra Tatarevic[2], Pavlos Kollias[1,3], Dié Wang[3], Kwangmin Yu[4], Andrew M. Vogelmann[3]

1. School of Marine and Atmospheric Sciences, Stony Brook University, Stony Brook, 11794, USA
2. Department of Atmospheric and Oceanic Sciences, McGill University, Montreal, H3A 0G4, Canada
3. Environmental and Climate Sciences Department, Brookhaven National Laboratory, Upton, 11973, USA
4. Computational Science Initiative, Brookhaven National Laboratory, Upton, 11973, USA

*Correspondence to*: Mariko Oue (mariko.oue@stonybrook.edu)

**Abstract.** Ground-based observatories use multi-sensor observations to characterize cloud and precipitation properties. One of the challenges is how to design strategies to best use these observations to understand these properties and evaluate weather and climate models. This paper introduces the Cloud resolving model Radar SIMulator (CR-SIM), which uses output from high-resolution cloud resolving models (CRMs) to emulate multi-wavelength, zenith-pointing, and scanning radar observables and multi-sensor (radar and lidar) products. CR-SIM allows direct comparison between an atmospheric model simulation and remote-sensing products using a forward-modeling framework consistent with the microphysical assumptions used in the atmospheric model. CR-SIM has the flexibility to easily incorporate additional microphysical modules, such as microphysical schemes and scattering calculations, and expand the applications to simulate multi-sensor retrieval products. In this paper, we present several applications of CR-SIM for evaluating the representativeness of cloud microphysics and dynamics in a CRM, quantifying uncertainties in radar-lidar integrated cloud products and multi-Doppler wind retrievals, and optimizing radar sampling strategy using observing system simulation experiments. These applications demonstrate the application of CR-SIM as a virtual observatory operator on high-resolution model output for a consistent comparison between model results and observations to aid interpretation of the differences and improve understanding of the representativeness errors due to the sampling limitations of the ground-based measurements. CR-SIM is licensed under the GNU GPL package and both the software and the user guide are publicly available to the scientific community.

## 1 Introduction

Ground-based observatories offer an integrated view of cloud and precipitation systems complementary to that available from satellites with excellent vertical resolution, especially in the boundary layer, and an accompanying description of the large-scale forcing. Currently, a number of observatories are continuously operated in different climate regimes (Illingworth et al., 2007; Löhnert et al., 2015; Stevens et al., 2016; Mather et al., 2016) with evolving measurement capabilities. In the beginning, zenith-pointing cloud radars, lidars, and radiometers provided the primary cloud and precipitation measurements.

Recently, the need to characterize the mesoscale organization of clouds and precipitation over a larger domain has heightened the sophistication and complexity of these observatories to go beyond single, one-dimensional profiling measurements. For example, the U.S. Department of Energy (DOE) Atmospheric Radiation Measurement (ARM) observatories offer observations from distributed networks of profiling and scanning radars, lidars, and radiometers (Turner and Ellingson, 2016; North et al., 2017).

Multi-parametric information from profiling and scanning radars, lidars, and radiometers has been used to retrieve cloud microphysical and kinematic properties, such as ice particle properties (e.g., Zhang et al. 2014; Kneifel et al. 2015; 2016; Matrosov et al. 2017; Von Lerber et al., 2017) and vertical velocities (North et al., 2017; Giangrande et al., 2016). However, the comparison between the retrieved observables (e.g., ice water content from radar reflectivity) and model-produced parameters often involves large uncertainties. Several factors, not limited to the nature of ground-based observations, challenge model evaluation using these observations. In many cases, the retrieval algorithms are based on statistical estimation of ill-posed inverse problems, and the results may not capture well the observed variability of natural data because of limitations from assumptions embedded in the retrieval algorithms (e.g., Szyrmer et al., 2012; Szyrmer and Zawadzki, 2014). Furthermore, determining critical parameters for model evaluation such as the cloud fraction profile requires complimentary, synergistic observations from both radar and lidar. One such example is the Active Remotely Sensed Cloud Location (ARSCL, Clothiaux et al., 2001) product that combines radar and lidar data to determine the hydrometeor height distributions. Other examples of critical parameters that require a multi-sensor approach include cloud and precipitation classification schemes (Illingworth et al., 2007) and hydrometeor phase classification (e.g., Shupe, 2007; Luke et al., 2010; Lamer et al., 2018). So, how do we best compare such products developed using multiple sensors with different capabilities (i.e., sensitivity) with numerical model output? Further, challenges may arise from the sampling strategy used to obtain the observations. For example, a recent study by Oue et al. (2016) has shown that zenith-pointing observations from one location are inadequate to provide reliable cloud fraction profile estimates in a cumulus field. A similar investigation on 3D wind retrievals in deep convection using multi-Doppler radar techniques highlights similar deficiencies of our current observing systems (Oue et al., 2019a). How do we best quantify the measurement uncertainty introduced by the observational strategy?

In this paper, we introduce the Cloud Resolving Model (CRM) Radar Simulator (CR-SIM), which has been continuously developed over the last five years to facilitate model-observation comparisons. CR-SIM applies forward simulators to atmospheric model output to simulate ground-based measurements. These simulations may be used: (1) to compare directly to the measurements, which provides an apples-to-apples comparison of the observed variables, or (2) as input to retrieval algorithms to assess the retrieval methodology or sampling strategy using the original atmospheric model output as 'truth'. In this study, the CR-SIM architecture and capabilities are presented along with a series of forward simulations that emphasize its capabilities. In particular, we highlight the applications of CR-SIM in investigations of observational uncertainties. Although accurate estimation of uncertainties in the retrieval products (e.g., ice water content, liquid water content, vertical velocity) is challenging, forward simulators allow us to emulate the observational retrieval products accounting for known error sources to understand the exact impacts of those error sources on the products by comparisons with the 'truth', which is

usually the input model data. Observing system simulation experiments (OSSEs) take advantage of forward modelling to produce simulated measurements. The understanding from OSSEs help: i) evaluate the model simulations using the observations while accounting for the observation limitations, ii) estimate uncertainties in retrieval techniques used, iii) propose new retrieval techniques accounting for the uncertainties, and iv) optimize new observation system strategies. This study demonstrates the application of the CR-SIM forward simulator in several OSSEs in which ARM multi-sensor products,

such as cloud locations and vertical velocity, are evaluated by considering limitations inherently imposed by the nature of the observations. (A list of acronyms is provided after the Appendix for easy reference.)

## 2. Forward Simulators

Forward simulators have been widely used to design observing systems and to provide an alternative path to model-observation comparisons by transforming the model geophysical quantities into remote sensing observables. There are several sophisticated radar simulators which have been developed for specialized applications of interest. For example, Snyder et al. (2017a, 2017b) simulate polarimetric radar characteristics of a supercell using radar forward simulators to understand the contribution of microphysical characteristics to the polarimetric properties and their wavelength dependency. They account

for the water fraction of solid ice particles to realistically simulate differential reflectivity ($Z_{DR}$) columns, specific differential phase ($K_{DP}$) columns, and co-polar correlation coefficient ($\rho_{hv}$) rings in supercells. A cloud radar simulator developed by Zhang et al. (2018) is designed to simulate vertically pointing cloud radar reflectivity (e.g., Ka- and W-band radars) using global climate model (GCM) output, which is beneficial for comparison of datasets at different scales (cloud-scale observational data versus global-scale data). Matsui et al. (2019) simulate polarimetric precipitation radar-based hydrometeor classification,

vertical velocity, and rain rate from CRM output to examine uncertainties in the retrieval algorithms and model microphysical parameterizations using the POLArimetric Radar Retrieval and Instrument Simulator (POLARRIS). The uncertainties are attributed to assumptions of hydrometeor particle size distribution, density, axis ratio, and/or canting angle. Lamer et al. (2018) developed a GCM-oriented ground-observation forward-simulator ((GO)$^2$-SIM), a comprehensive radar-lidar simulator for the GCMs that emulates radar Doppler spectra moments, lidar backscatter and depolarization, and provides synthetic estimates of

mixed-phased cloud occurrence in the GCMs that are comparable to those estimated from observations using the same methodology.

CR-SIM can simulate the ideal multi-wavelength radar and lidar observables, and multi-sensor integrated products. The zenith-pointing and scanning radar observables include radar reflectivity, Doppler velocity, spectrum width, and polarimetric fields. Zenith-pointing lidar observables include lidar backscatter and extinction coefficient. The idea behind CR-SIM is to

have a forward model operator that provides *idealized* radar and lidar observables (i.e., actual observations after perfect quality control and correction for the total (two-way) attenuation) on the same grid as in the CRMs or large-eddy simulations (LESs) to facilitate model-observation comparisons. Further, the design is flexible enough to be coupled with different microphysical

schemes and different scattering methods (e.g., T-matrix, Mishchenko, 2000; Discrete Dipole Approximation, Yurkin and Hoekstra, 2011).

The CR-SIM forward simulator is tailored to compute radar and lidar observables by integrating scattering properties over the discrete particle size distributions (PSDs) using a constant bin size for each hydrometeor (Table 1), based on the microphysical scheme used in the CRM or LES. The environmental variables are obtained from a mandatory set of model output variables consisting of pressure, temperature, dry air density, and height above sea level. The single-scattering properties are calculated using the T-matrix method and packaged as look-up tables (LUTs) in CR-SIM. The simulated idealized radar

and lidar variables are provided at each model grid box and can be easily compared with real observations.

## 2.1. Scattering Properties

    The LUTs consist of the complex scattering amplitudes $S_{ij}$ of the $2 \times 2$ scattering matrix for single, non-spherical particles

at fixed orientations with equally spaced particle sizes computed using the T-matrix code of Mishchenko and Travis (1998) and Mishchenko (2000). Following Ryzhkov et al. (2011), we assume that the scattering characteristics of arbitrarily oriented particles can be expressed via the scattering amplitudes $f_a$ and $f_b$ corresponding to the principal axes of spheroid when the electric vector of the illuminating electromagnetic wave is directed along the spheroid axes $a$ and $b$ respectively. Each hydrometeor specie is characterized by its equivalent volume diameter, $D=(ab^2)^{1/3}$, where $a$ is the symmetry axis of the spheroid

and $b$ is its transverse axis. In this convection, an oblate hydrometeor has $a < b$. The scattering amplitudes $f_a$ and $f_b$ correspond to $S_{vv}$ and $-S_{hh}$ computed by the Mishchenko (2000) T-matrix code, respectively ($f_a \equiv S_{vv}$; $f_b \equiv -S_{hh}$). The minus sign in expression for $f_b$ is to account for switching from the forward-scattering convention used for the amplitude and phase matrix components in the T-matrix code to the backscatter-alignment convention used in the definitions of polarimetric radar measurements except for forward-scattering parameters (attenuation and phase shift).

The LUTs of scattering properties for single particles are constructed for each hydrometeor class corresponding to the CRM or LES simulation output (e.g., rain drop, snowflakes, cloud droplet, ice crystal, graupel) as a function of particle phase, bulk density, aspect ratio, size, and temperature. We used fixed values of these parameters depending on the "scattering type," which must be assigned to the corresponding hydrometeor class. The bulk density used is as parameterized in the selected microphysics scheme, assuming spheroid particle shapes. For each hydrometeor class, the complex scattering amplitudes are

calculated for 91 elevation angles from 0° to 90° with a spacing of 1°, five radar frequencies at 3 GHz (S band), 5.5 GHz (C band), 9.5 GHz (X band), 35 GHz (Ka band), and 94 GHz (W band), different temperature ranges for the liquid hydrometeors, different particle densities for solid hydrometeors, and few different values of particle aspect ratio. The scattering types built in the current LUTs and their parameter settings are presented in Table 2. For lidar scattering properties, the single-particle extinction $\sigma_\alpha$ and backscattering cross section $\sigma_\beta$ for spherical cloud droplets and cloud ice are calculated using the BHMIE

Mie code (Bohren and Huffman, 1998). CR-SIM operates for observables from the ceilometer (wavelength of 905 nm) and micro pulse lidar (MPL, wavelengths of 353 and 532 nm).

Although most of the parameters related to hydrometeor particles (e.g., particle bulk density, size) required in the scattering calculations can be obtained from the prognostic and diagnostic variables from the CRM or LES, aspect ratios and canting angles must be assumed in the simulator and as such are prescribed by the users. All liquid and ice hydrometeors are assumed to be oblate spheroids, except cloud droplets. Raindrops are represented as oblate spheroids with a size-dependent aspect ratio following an empirical equation as a function of particle diameter, based either on Brandes et al. (2002) or Andsager et al. (1999). A fixed aspect ratio is used for each solid hydrometeor category, but for graupel and hail the empirical expression proposed by Ryzhkov et al. (2011) is also available. For all model hydrometeors, the radar polarimetric variables (which depend on particle orientation) are calculated using complex scattering amplitudes from the pre-built LUTs, assuming a mean particle canting angle of 0° (Ryzhkov 2001) with a choice of the particle orientation distribution. The possible choices for the distribution of particle orientations are: fully (three-dimensional) random orientation, random orientation in the horizontal plane, and a two-dimensional axisymmetric Gaussian distribution of orientations. In this paper, for all simulations, we used aspect ratios proposed by Brandes et al. (2002) for rain drops, 0.2 for cloud ice, 0.6 for snow, Ryzhkov et al. (2011) for graupel and hail, and the two-dimensional axisymmetric Gaussian distribution for all hydrometeor species.

A hydrometer in CR-SIM is either pure liquid or a mixture of ice and air. The composition of particles within a volume is represented in the scattering computations by an appropriate selection of the dielectric constant for different hydrometeor types. The dielectric constant of liquid particles is frequency- and temperature-dependent (Ray, 1972). Ice phase hydrometeors are assumed to be composed of ice and air in an ice matrix and their effective dielectric constant is computed using the Maxwell-Garnet mixing formula (Maxwell Garnet, 1904). The output radar reflectivity ($Z_{hh}$) for all hydrometeor species is the equivalent radar reflectivity, for which the computations adopt a dielectric factor of 0.92 for all hydrometeor species. This choice of the dielectric factor ensures a convention that the definition of radar reflectivity reduces to form: $Z = \int N(D)D^6 dD$ for (spherical) liquid particles, where $D$ is the droplet diameter and $N(D)$ is the droplet size distribution function.

The LUTs of scattering properties currently incorporated in CR-SIM were created using the T-matrix method where solid phase hydrometeors are represented as dielectrically dry oblate spheroids. Although these assumptions are rather simple compared to some other radar simulators that take into account complex electromagnetic scattering by mixed-phase hydrometeors or ice hydrometeors with possibly irregular shapes are taken into account (e.g., Snyder et al., 2017a,b; Jiang et al., 2019), such complex electromagnetic scatters can be easily incorporated by adding LUTs of their scattering properties from different scattering calculation methods (e.g., Kneifel et al., 2017; Leinonen and Moisseev, 2015; Leinonen and Szyrmer, 2015; Lu et al., 2016) without any structural change to the code.

### 2.2. Calculations of radar and lidar observables

The PSD for each hydrometeor specie is produced based on the model microphysics scheme. The incorporated microphysics schemes and corresponding CRM currently available in CR-SIM are listed in Table 3. CRMs coupled with bulk moment microphysics (i.e., single and double moment) prognose mixing ratio and/or the number concentration of each

hydrometeor specie. Using these parameters, the PSD is determined in combination with size distribution assumptions used in the microphysics scheme. Bin microphysics schemes explicitly calculate the evolution of the PSDs. Radar moment observables are computed by integrating scattering properties (see the Appendix) from the LUTs over the discrete PSD for each hydrometeor type following Ryzhkov et al. (2011) while accounting for an orientation distribution as described in section 2.1,

and then integrated over all simulated hydrometeor species to produce a unique value for each observable at each grid box (see Tatarevic et al., 2019 for the detailed descriptions). Computed radar variables are listed in Table 4.

Particle fall velocity, which is used for Doppler velocity and spectrum width computations, is parameterized as a function of particle diameter in the same manner as in the selected microphysics scheme. The fall velocity size relationship ($V_f(D)$) for each hydrometeor specie is specified in a form:

$$V_f(D) = f_c a_v D^{b_v} \qquad\qquad (1)$$

where $f_c = (\rho_{surf}/\rho)^k$ is the correction factor for air density with exponent $k$. The values for the prefactor $a_v$, the exponent $b_v$, and the exponent $k$ vary according to the microphysics scheme and do not depend on particle orientation. The air density at sea level, $\rho_{surf}$, is computed for the first model level. The values of coefficients and detailed descriptions concerning the microphysics schemes are found in Tatarevic et al. (2019).

The versatility of the CR-SIM computational capabilities is briefly demonstrated by the following examples.

- Figure 1 shows an example of S-band (3 GHz) radar observables from CR-SIM for a mesoscale convective system (MCS) observed on May 20, 2011, during the Midlatitude Continental Convective Clouds Experiment (MC3E; Jensen et al., 2016). The convective system was simulated using the Weather Research Forecasting (WRF) model (Skamarock et al., 2008) with the Morrison 2-moment microphysics scheme, a horizontal

resolution of 0.5 km, and the vertical resolution of approximately 0.25 km.

- CR-SIM includes a computation of the Doppler power spectra by introducing the method used in Kollias et al. (2014). Figure 2 shows examples of the Doppler spectra and its moments at S-band for a WRF-simulated cumulus field. In the figure, a pulse repetition frequency (PRF) of 600 Hz is used, the noise power at 1 km is -40 dB, and the number of Doppler velocity bins is 256.

- CR-SIM also includes forward simulators for the ceilometer (wavelength of 905 nm) and ground-based micro pulse lidar (wavelengths of 532 and 353 nm). The lidar observables are computed for cloud ice and cloud droplet species (see Table 5 and the Appendix). Figure 3 shows an example of profiles calculated for lidar observables for a cumulus case from the LES ARM Symbiotic Simulation and Observation project (LASSO, Gustafson et al., 2020) using WRF coupled with the Morrison 2-moment microphysics scheme. In this simulation, typical

profiles are presented for aerosol backscatter ($\beta_{aero}$) and extinction coefficient ($\alpha_{ext\_aero}$), and molecular backscatter ($\beta_{mol}$) based on Spinhirne (1993). As expected, the lidar backscatter is significantly attenuated by cloud droplets, but the very high lidar backscatter at the interface between air and cloud can be used to detect cloud base height.

## 2.3 Instrument model

An instrument model is used to post-process the CR-SIM results to account for the total (two-way) attenuation in a pathway and the effects of technological specifications on the observables, such as sampling volume and detector sensitivity. The standard output of CR-SIM consists of synthetic profiling radar and lidar observations at each grid box of the model domain, and synthetic scanning radar observations for a radar positioned at a user-specified location inside the model domain. The output synthetic fields are artifact-free, with no propagation (e.g., total attenuation in a pathway) or instrument sampling effects (e.g., antenna beamwidth, range-gate spacing). This approach is based on the notion that the real observations used for comparison against the synthetic simulated observables will have undergone rigorous post-processing that mitigate attenuation effects to the extent possible, velocity folding etc. However, the user can emulate the true observations of a scanning radar while selecting the placement of a radar or a network of radars within the model domain, thus, imposing a volume coverage pattern (VCP) scan strategy. In this case, the idealized, standard CR-SIM output at the model grid resolution can be further used as input into a radar instrument model that is written specifically for the post-processing of the CR-SIM radar simulations. The radar instrument model accounts for radar distance to the target, elevation as provided by the VCP, pulse length, range resolution, antenna beamwidth, and receiver noise and output the radar observables in radar polar coordinates. For the calculation of elevation angles, the earth surface is assumed to be flat, which is an acceptable assumption for general radar observation ranges ($<$ ~90 km for the vertical model grid spacing of $>$ 0.5 km). Gaussian functions are used as the antenna-weighting and the range-weighting functions to estimate the contribution of the model grid observables to the radar polar coordinate system observables. Depending on the azimuthal resolution and the antenna beamwidth, this instrument model also accounts for the radar sampling resolution.

Radar reflectivity attenuated for hydrometeors can be computed using the integrated specific attenuation for hydrometeors along a radar beam path. The total hydrometeor attenuation ($A_{tot}$) at each grid box is then equal to twice the sum of the specific attenuation for all simulated hydrometeors ($A_h$) along a radar beam path from the location of the radar to a distance of the target at $r$ in km:

$$A_{tot}(r) = 2 \int_0^r A_h(r)\, dr \qquad (2)$$

where $A_{tot}$ is in dB and $A_h$ in dB km$^{-1}$. Here gaseous attenuation was not included. The observed reflectivity $Z_{hh}{}^{obs}$ (logarithmic scale) is computed by subtracting $A_{tot}$ from $Z_{hh}$ on a logarithmic scale:

$$Z_{hh}{}^{obs}(r) = Z_{hh}(r) - A_{tot}(r) \qquad (3)$$

Like $Z_{hh}$, the attenuated differential reflectivity $Z_{DR}{}^{obs}$ on a logarithmic scale is calculated as:

$$Z_{DR}{}^{obs}(r) = Z_{DR}(r) - 2 \int_0^r A_{dp}(r) \, dr \qquad (4)$$

235

where $A_{dp}$ represents specific differential attenuation in dB km$^{-1}$. The minimum detectable reflectivity $Z_{MIN}$ (logarithmic) is applied with a constant $C$:

$$Z_{MIN} = C + 20 \, log_{10}(r) \qquad (5)$$

240

where $r$ is the radial distance in km, and the constant $C$ represents the minimum detectable signal at $r = 1$ km for the pulse length selected by the user.

Figure 4 shows simulated range-height indicator (RHI) measurements at C and X bands (5.5 and 9 GHz, respectively) accounting for $Z_{MIN}$, total hydrometeor attenuation, and the radar range-gate sampling volume for convective cells associated with an MCS observed on May 20, 2011 during MC3E. The input convective system simulation data are the same as Figure 1. The instrument specifications used for the RHI simulations are for the X-band radar, a beamwidth of 1°, range-gate spacing of 50 m, and a constant $C$ of -50 dBZ for the $Z_{MIN}$ calculation. The C-band radar specifications are a beamwidth of 1°, range-gate spacing of 120 m, and a constant $C$ of -35 dBZ for the $Z_{MIN}$ calculation. These specifications follow the X-band scanning ARM precipitation radar (X-SAPR) and C-band scanning ARM precipitation radar (C-SAPR) configurations at the ARM Southern Great Plains (SGP) site during MC3E. The results are reasonable, showing strong attenuation in $Z_{hh}$ and $Z_{DR}$ by rain at X band and relatively less at C band. The simulated $K_{DP}$ at X band is approximately 1.6 times larger than that at C band because of the wavelength dependency.

## 2.4. Code Features

CR-SIM is written in Fortran 95 standard including all GNU extensions and parallelized with OpenMP. The input to CR-SIM is the output from a CRM or LES in NetCDF format. The output of CR-SIM is in NetCDF format and includes simulated observables for each hydrometeor specie, and one total for all the hydrometeors. These features allow users to understand the contributions of each hydrometeor specie to the radar observables for a sophisticated evaluation of microphysics schemes. The code includes various microphysics schemes as shown in Table 3. The code structure supports different CRMs or LESs, flexible microphysics package extensions, and diverse assumptions such as particle shape, density, and PSD for different hydrometeor categories in the models as well as inclusion of scattering properties computed by different methods.

The CR-SIM runtime depends on the computing power, number of threads used, simulation domain size, and the number of cloudy grid boxes. The simulations presented in this manuscript were run on a computer having 500 GB memory and 24

processors each with 12 cores. For the MCS case in Fig. 1, having a simulation domain size of $667 \times 667 \times 12$ (5.3 M grid points), the runtime is approximately 270 seconds using 16 threads.

The code has been released under GNU General Public License and both the software and a detailed user guide are publicly available online (Tatarevic et al., 2019).

## 3. Sample Applications of CR-SIM

In this section, several applications of CR-SIM are presented that highlight its capabilities. These applications are: i) a comparison of observed and modeled cloud fraction profiles (CFPs); ii) a quantification of uncertainty in the estimate of domain-averaged CFP; iii) an evaluation of a novel retrieval technique for the estimation of cloud fraction; iv) an investigation of the impacts of limitations imposed by the nature of observations themselves on multi-Doppler wind retrieval techniques; and v) an optimization of a new radar observation strategy for multi-Doppler wind retrievals.

Figure 5 shows a flow diagram of our application processes. First, the forward simulator produces idealized observables at each model grid box (the 'Output 1' box in Figure 5). In the second step, an instrument model is used to account for the instrument features (as described in section 2.3). Third, the output from the instrument model ('Output 2') is then used to retrieve the CFP (the retrieval model and 'Output 3') for a direct comparison and, most importantly, for a quantification of the uncertainties in the cloud fraction estimates and evaluation of the new retrieval technique (applications i – iii). On another hand, the output from the instrument model is also used as an input for multi-Doppler wind retrieval model to investigate the uncertainty of the retrieval method and to optimize the new radar observation strategy (applications iv and v, with 'Output 3'). The final step consists of a comparison of the retrieved quantities using a multi-Doppler wind retrieval against the input CRM or LES data, and a quantitative estimation of uncertainties attributed to the observation limitations and the retrieval algorithms. In the following subsections, we briefly describe and summarize the findings of the studies using CR-SIM.

### 3.1 Comparison of observed and modeled cloud fraction profiles

Measurements of the CFP are important to quantify the impact of shallow cumulus clouds on the grid-scale meteorological state because the fractional cloudiness of a grid box has an impact on the radiative transfer (e.g., Albrecht 1981; Larson et al., 2001) and the vertical cumulus mass flux (e.g., de Roode and Bretheton, 2003; van Stratum et al., 2014). Zenith-profiling cloud radar and lidar measurements traditionally have been used to provide CFP estimates (e.g., Hogan et al., 2001; Kollias et al., 2009; Remillard et al., 2013; Angevine et al., 2018). Typically, the profiling radar and lidar observations are combined synergistically to provide a hydrometeor mask such as those described in ARSCL (Clothiaux et al., 2000) and the CloudNet target classification (Illingworth et al., 2007). This approach takes advantage of the radar and lidar capabilities and maximizes our ability to detect thin cloud layers. However, the performance of the combined radar/lidar algorithm degrades at heights

where the lidar observations are unavailable due to complete signal attenuation. These attenuation effects are naturally not represented in model output and thus may lead to large disagreements between observations and models.

Our focus is to use CR-SIM to generate a synthetic ARSCL product that is directly comparable to the ARSCL generated using measurements from the Ka-band ARM Zenith-pointing Radar (KAZR), ceilometer, and MPL. This analysis uses a shallow cumulus cloud field over ARM SGP site simulated by the LASSO project. The simulation is for the June 27, 2015 case and uses WRF-LES coupled with the Morrison double moment microphysics scheme (Morrison et al., 2005). The horizontal and vertical resolutions are 100 m and 20 m, respectively, and the horizontal dimension of the simulation domain is 14.4 km.

First, the KAZR, ceilometer, and MPL measurements from the ARM SGP Central Facility are simulated using the CR-SIM forward simulator. The simulation output corresponds to the box 'Output 2' in Figure 5. Simulated KAZR reflectivity accounts for hydrometeor attenuation ($Z_{hh}{}^{obs}$) and radar sensitivity ($Z_{MIN}$) as described by Eqs. (2, 3, and 5). The attenuated MPL hydrometeor backscatter ($\beta_{hydro\_atten}$) is obtained by subtracting $\beta_{aero\_atten}$ and $\beta_{mol\_atten}$ from $\beta_{total\_atten}$, since the MPL total backscatter includes aerosol backscatter and molecular backscatter (see Table 5). The value obtained for $\beta_{hydro\_atten}$ is below noise level if less than the unattenuated background scatter ($\beta_{aero} + \beta_{mol}$), which is used in this simulation as the minimum detectable MPL backscatter value. The ceilometer-detected first cloud base is estimated at each grid column following O'Connor et al. (2004). Using the simulated observables, we estimate cloud locations as provided by ARSCL ('Output 3' in Fig. 5). A grid box where either KAZR $Z_{hh}{}^{obs}$ or MPL $\beta_{hydro\_atten}$ has a detectable value is indexed as a 'cloudy' grid box, and grid boxes below the simulated ceilometer first-cloud base are indexed as 'clear'.

An example of an ARSCL simulation is shown in Fig. 6 that uses the LASSO LES data as an input. The WRF simulation shows cumulus clouds below 5 km and cirrus clouds covering the entire domain at 12-14 km. In Figs. 6b-d, the limitation of each instrument is represented in the forward simulations. The simulated KAZR measurements can detect cumulus cloud layers but cannot detect cirrus clouds, due to their low reflectivity value (lower than $Z_{MIN}$). Instead, the cirrus clouds can be detected by the simulated MPL measurements. However, the cirrus clouds may be missed by both radar and lidar measurements when cumulus clouds are present, because the MPL signal becomes fully attenuated by the low-level clouds. Figures 6f and 6g show the domain-averaged CFPs from the LES hydrometeor mixing ratio and from the simulated ARSCL which assumes the ARM instruments are located at every grid column (as shown in Fig. 6e). Comparison between the two CFP plots suggests that the ARSCL for this LASSO case underestimates cirrus CFPs by 20%, likely due to lidar beam attenuation by lower-level cumulus clouds that have a horizontal fraction of 20%.

### 3.2 Uncertainty quantification of domain-averaged cloud fraction profile estimates

The ARSCL product is usually integrated for 1-3 hours to provide an average CFP estimate for that time period. These CFP estimates are often compared with the model domain-averaged CFPs. However, the spatially heterogeneous distribution of the shallow cumulus clouds (Wood and Field, 2011) raises questions regarding the ability of short-term (1–3 hours) zenith-

profiling observations to provide adequate sampling of the cloud field. Uncertainties in the profiling measurement of cloud fractions are introduced by the limited sampling of a highly heterogeneous cloud field. We investigate these uncertainties as a function of the number of profiling sites and integration time using the CR-SIM virtual observations and the WRF LES simulation presented in the previous application. The WRF LES output is saved every 10 minutes, and CR-SIM is run for each

output file. In this analysis, we assume that no cloud evolution occurs within a 10-minute interval.

Figures 7a and 7b show the domain-averaged CFP from the simulated ARSCL and directly from the WRF LES using a cloud water content threshold of 0.01 g kg$^{-1}$. The colors indicate different integration periods. Note that the WRF dataset in this analysis is for a shallow convective case at SGP on June 11, 2016, different from the one used in Fig. 6, which has higher cumulus cloud tops. The simulated ARSCL CFP is in good agreement with the WRF CFP for each integrated period (compare

Figs. 7a and b), indicating that uncertainties attributed to observation limitations (e.g., sensitivity and attenuation) are small. Thus, the limited spatial and/or temporal sampling is the major error source to consider when comparing the profiling measurement derived CFP with the domain-averaged WRF CFP.

To emulate vertical profiling measurements, we sampled data as follows. First, observation sites are randomly selected within the horizontal domain. Second, for each snapshot of the simulation, clouds over the observation sites are sampled as if

the clouds are frozen in time and advected by the mean environmental wind. Thus, the columns are sampled along the direction of the horizontal wind over the advected distance (i.e. horizontal wind speed × 10 min), where the environmental horizontal wind at each snapshot is the mean horizontal wind across the simulation domain within the cumulus cloud layer (i.e., the layer between the mean cumulus cloud base and the maximum cumulus cloud top). Last, the CFP is estimated by varying both the number of observation sites and the integration period.

Figures 7c-h show the comparisons of the WRF CFPs and the simulated ARSCL CFPs for different number of observation sites (top row, c-e) and integration periods (bottom row, f-h). The center of the integration period is 21:00 UTC. Blue lines in each panel represent the simulated ARSCL CFPs integrated over time from each selected observation site for the period indicated, the red line represents the mean ARSCL CFP averaged over the sites, and the black line represents the domain-averaged WRF CFP integrated over the indicated period. Each panel shows that the CFPs at a single site (blue lines) have large

uncertainties even though they are integrated over long periods, ranging from 30 to 180 min. Those uncertainties are reduced when averaging the CFP profiles across the different sites; consequently, the mean CFP (red line) becomes closer to the domain-averaged WRF CFP (black lines). However, it also becomes evident that a small number of observation sites (Fig. 7c) may not be adequate to estimate the true CFP.

Figures 7i and 7j show the root mean square error (RMSE) and mean absolute percentage error (MAPE) of the simulated

ARSCL CFPs as a function of the number of observation sites and the integration time. Both plots show that the uncertainty can be reduced by increasing the number of observation sites and the length of the integration period. The RMSE dramatically decreases to 0.005-0.01 (30-50 % in MAPE) when we use four observation sites and 120 min integration. The rate of improvement of CFP by further increasing the number of sites and integration period is smaller; the error values slowly decrease until the RMSE and MAPE plateau at 0.002 and 15%, respectively. However, establishing more than ten observational

sites in such small domain is probably impractical. At the SGP site, five Doppler Lidar profiling measurements have already been operating over a 90 km × 90 km domain. These measurements can be effectively used to estimate cloud fraction without much uncertainty when clouds are homogeneously distributed over the domain.

### 3.3 Evaluation of a new CFP estimation technique using scanning cloud radar


Forward-radar simulators can be used to evaluate retrieval techniques. We introduce an application to estimating CFP using scanning cloud radar (SCR) measurements presented in Oue et al. (2016). As analyzed in the previous section, profiling radar measurements may produce large uncertainties in CFP estimates. On the other hand, SCRs conduct observations over a much larger domain than zenith-profiling cloud radars such as KAZR (e.g., Lamer et al., 2013; Ewald et al., 2015). Although

the SCRs are widely and routinely used to observe 3D cloud fields, the application of SCRs to study shallow cumuli is not straightforward. One of the most significant limitations of the SCR observations is related to the radar sensitivity. Since shallow cumuli over land typically have low reflectivities, the strong reduction in SCR sensitivity with range creates the illusion of the atmosphere being cloudier closer to the radar location (e.g., Lamer and Kollias, 2015). This limitation can introduce uncertainties in the cloud fraction estimates. Oue et al. (2016) addressed uncertainties of radar-estimated CFPs due to the

nature of the profiling and scanning radar techniques using CR-SIM-generated observations.

Figure 8a shows horizontal cross sections of WRF-simulated water content for a shallow convection case (June 9, 2015; Oue et al., 2016) from LASSO. Figure 8b shows the CR-SIM simulation of the Ka-band (35 GHz) $Z_{hh}$ accounting for the minimum detectable reflectivity $Z_{MIN}$ of the cross-wind RHI (CWRHI, Kollias et al., 2014) scans from Eq. (5) using C=-50 dBZ. In the CR-SIM analysis, the radar was located along the vertical line in Figure 8b, and CWRHI scans were performed

along the east-west direction while the clouds were assumed to move along the north-south direction. These figures suggest that the $Z_{hh}$ from the CWRHI scans cannot capture the clouds with lower water contents that are located far from the radar. This can affect cloud fraction estimates. Because the "true" cloud fraction is estimated from the original model cloud field and thus is known, the CR-SIM runs in different configurations can be used to establish the best method to estimate the cloud fraction while accounting for limitations inherent to the nature of radar measurements. Oue et al. (2016) use the cumulative

distribution function (CDF) of the observed $Z_{hh}$ to define the size of the horizontal domain at each height needed to obtain the best estimate of the domain-averaged CFP. The horizontal domain size as a function of height corresponds to a distance from the radar where $Z_{MIN}$ was equal to a CDF value of 10%. Figure 8c shows CFPs using a CDF of 10% when changing the integration time of the CWRHI, and Figure 8d shows the RMSE of the estimated CFPs as a function of integration time, adapted from Oue et al. (2016). The figure suggests that the 35 min or longer of CWRHI measurements provide the realistic

domain-averaged CFP.

### 3.4 Investigation of impacts of observation limitations on multi-Doppler radar wind retrievals

Estimation of vertical air motion is essential to understand the dynamics and microphysics of deep convective clouds (e.g., Jorgensen and LeMone, 1989; Wang et al., 2019), evaluate CRM and LES results (e.g., Varble et al., 2014; Fan et al., 2017), and improve convective parameterization in GCMs (e.g., Donner et al., 2001). Multi-Doppler radar techniques have been applied to understand the dynamics and microphysics of the deep convective clouds in different climate regimes (e.g., Friedrich and Hagen, 2004; Collis et al., 2013; Oue et al., 2014). However, the multi-Doppler radar retrievals are not straightforward with potential uncertainties from multiple aspects (e.g., Clark et al., 1980; Bousquet et al., 2008; Potvin et al., 2012). CR-SIM can be used to investigate the impacts of different error sources on the retrieved wind fields.

Oue et al. (2019a) investigated the impacts of the radar VCP for the plan position indicator (PPI) and the observation period on uncertainties in multi-Doppler radar wind retrievals using CR-SIM. They also investigated how the uncertainties attributed to the VCP period can be reduced using the advection-correction technique proposed by Shapiro et al. (2010). The advection correction scheme allows for trajectories of multiple individual clouds, performs smooth grid box-by-grid box corrections of cloud locations, and takes into account changes in cloud shape with time by using PPI scans at two times. We summarize their findings, particularly regarding the impacts of radar VCP and period on multi-Doppler radar retrievals.

Figure 9 shows a diagram of the analysis process. The input model data is a WRF simulation using the Morrison double-moment microphysics scheme for a MCS observed on May 20, 2011, during the MC3E field campaign at the ARM SGP site. The horizontal resolution is 500 m, the vertical resolution varies from approximately 30 m near the surface to 260 m at 2 km—above which the resolution remains approximately constant, and the simulation output is saved every 20 seconds. Measurements from the three X-band scanning ARM precipitation radars (X-SAPR) at the SGP site are simulated using CR-SIM. The CR-SIM-simulated radar reflectivity and Doppler velocity at the model grid are converted into the radar polar coordinates with two different VCPs for each radar: 1) 21 elevation angles ranging from 0.5° to 45° (VCP1, same as the X-SAPR scan strategy during MC3E), and 2) 60 elevation angles ranging from 0.5° to 59.5° with a 1° increment (VCP2). For the both VCPs, the beamwidth is 1°, the range-gate spacing is 50 m, and the maximum range is 40 km. The simulated radar reflectivity and Doppler velocity in polar coordinates were used as an input to the 3DVAR multi-Doppler radar wind retrieval algorithm developed by North et al. (2017) to estimate the 3D wind field for a domain of 50 km × 50 km × 10 km with horizontal and vertical grid spacings of 0.25 km.

The convective mass flux (MF) is estimated at each height as:

$$MF = UF \; \overline{w} \; \overline{\rho_d} \quad [kg \; s^{-1} \; m^{-2}] \tag{5}$$

where $UF$ is updraft fraction over the horizontal slice of the domain, $\overline{w}$ is mean vertical velocity over the updraft area, and $\overline{\rho_d}$ is dry air density averaged over the domain. Figure 10 shows comparisons of convective mass flux profiles between simulated multi-Doppler radar retrievals and WRF output for two minimum updraft thresholds of 5 ($MF_5$) and 10 ($MF_{10}$) m s$^{-1}$. First, we applied the wind retrieval technique to a snapshot of the forward-model output to bypass the instrument model and examine

the uncertainty in the retrieval model (3FullGrid). Figure 10a shows MF profiles from the 3FullGrid simulation (red line) and from the WRF snapshot (black line), 2-min average (dark gray line), and 5-min average (light gray line). The 3FullGrid MF profile is in good agreement with the WRF output, indicating that the uncertainty in the retrieval model is small; although, it

does underestimate the maximum MF for the updraft threshold of 5 m s$^{-1}$ by 0.05 kg m$^{-2}$ s$^{-1}$ (10% of the true MF) at 5.3 km.

Figures 10b and 10c show MF profiles (MF$_5$ and MF$_{10}$) obtained from simulated retrievals while considering the effects of VCP (VCP1 and VCP2) and averaging period (snap [instantaneous], 2-min, and 5-min averages). For both VCP1 (Fig. 10b) and VCP2 (Fig. 10c), the snapshot and 2-min VCP simulations have similar MF estimates for both sets of MF$_5$ and MF$_{10}$ curves, indicating that a 2-min average is sufficient to capture features available from an instantaneous scan. However, the

accuracy of these estimates varies with MF profile and VCP. The MF$_{10}$ estimates for both VCPs systematically underestimate the maximum values occurring between 4.5-6.5 km by about 0.5 kg m$^{-2}$ s$^{-1}$ (20%). The performance of the MF$_5$ estimates for VCP snap and 2-min have strong variations with height. For VCP1 (the less dense scan pattern), MF$_5$ follows the WRF snapshot below 4.5 km with close agreement between 3-4.5 km; however, MF is underestimated around its maximum MF by about 0.075 kg m$^{-2}$ s$^{-1}$ (15%) and is overestimated below 3 km and above 7 km. The denser scan pattern for VCP2 provides a

dramatic improvement around the maximum and above 6 km, but it still shows overestimations below 3 km and above 7 km. Uncertainties are often increased for the VCP simulations when the averaging period is extended to 5-min. For the 5-min VCPs, MF$_{10}$ estimates for both VCP1 and VCP2 around the maximum are further underestimated while the MF$_5$ estimate for VCP2 is further overestimated above 6 km. Other estimates below this height for VCP2 and for all heights for VCP1 are mostly unchanged. These results suggest that the VCP elevation strategy and sampling time extended to 5 min have a significant

impact on the updraft properties retrieved at higher altitudes. This is due to density of data sampled by the VCPs, where greater density particularly improves MF$_5$ around its maximum, and the deformation of cloud structures within longer sampling periods (exceeding 2 min) that causes uncertainties in the mass continuity assumption.

The rapid volume scan of less than 5 minutes required in the retrieval of the high-quality vertical velocities is challenging for conventional scanning radars. Most of the improvements required in the sampling strategy of the observing system (higher

maximum elevation angle, higher density elevation angles, and rapid VCP time period) can be accomplished using rapid scan radar systems such as the Doppler on Wheels mobile radars (DOW; Wurman, 2001) or phased array radar systems (e.g. Kollias et al, 2018).

### 3.5 Evaluation of new radar observation strategies


CR-SIM can also be used to examine performances of new remote sensing systems and help to choose the most appropriate observation strategy for a new field campaign. Figures 4c and 4d show the performance of C-band (5.5 GHz) RHI measurements when the radar is located at 24 and 59 km away from the target convective clouds. As expected, the RHI from the greater distance provides the radar observables at lower resolution and includes more attenuation when precipitation clouds

are located between the target and the radar. Oue et al. (2019a) investigate the impact of radar data sampling on the multi-

Doppler radar wind retrievals for the MCS by an OSSE using CR-SIM. The addition of data from a Doppler radar to form a triple-Doppler radar retrieval, shown in section 3.4, cannot significantly improve the updraft retrievals if the added radar VCP has inferior spatial resolution. Oue et al. (2019a) also show that the updraft retrievals in a limited area around the center of the domain, where data density from the three radars are higher than other areas, produced better results than those in the entire

domain. The insights obtained from these OSSEs are beneficial for decision-making regarding radar observation strategies for a field campaign, such as the number of radars required and their locations. For example, Kollias et al. (2018) used CR-SIM to examine how phased array radars improve multi-Doppler radar wind retrievals compared to scanning radars for MCSs.

### 4. Summary


We present a recently developed comprehensive forward simulator for radar and lidar, CR-SIM, which is suitable for simulating complex, ground-based observational configurations and their synthetic products. CR-SIM can simulate multi-wavelength, zenith-pointing and scanning radar observables (radar reflectivity, Doppler velocity, polarimetric fields, radar Doppler spectrum), lidar observables, and multi-sensor integrated products. The primary idea behind the simulator is to directly

compare remote sensing observations with simulated measurements based on the CRM or LES output, maintaining consistency with the microphysics scheme used in the model. CR-SIM incorporates microphysical and scattering properties independently so that uncertainties related to microphysical assumptions are separated from uncertainties related to scattering model. This configuration allows CR-SIM to be easily expanded, either by adding microphysical schemes or new scattering classes.

One feature of CR-SIM is that it produces both radar and lidar observables for all the CRM grid boxes while accounting

for elevation angles relative to a radar location, similar to Snyder et al. (2017a,b). Other radar simulators also account for radar geometry characteristics such as beamwidth and radar range resolution to simulate scatterers within the radar resolution volume (e.g., Capsoni et al, 2001; Caumont et al., 2006; Cheong et al., 2008). Instead, here, the radar sampling characteristics (e.g. antenna beamwidth, range-bin spacing, total attenuation, sensitivity) are accounted for in our post-processing instrument model. This feature facilitates configuring any desirable observational setup with a varying number of profiling or scanning

sensors from a single model simulation. The CR-SIM multi-sensor simulations include multi-wavelength radars and lidars that allow simulation of sophisticated virtual products such as ARSCL and 3DVAR multi-Doppler based wind retrievals. The CR-SIM applications shown in this paper emphasize the value of applying it to high-resolution model output to emulate the sampling by the ground-based observatories. CR-SIM's coupling of CRM microphysical parameterizations with scattering models facilitates improved evaluation of model performance by enabling robust comparisons between model-simulated

clouds and observables from radar and lidar while accounting for instrument characteristics and observation limitations.

CR-SIM is easily expanded to include additional microphysical schemes, new scattering classes, scattering calculations, and other applications to simulate multi-sensor products (e.g., multi-Doppler wind retrievals, ARSCL). At this stage, all ice hydrometeors (e.g., snow, ice, graupel, hail) are modeled as dielectrically dry spheroids. The LUTs of scattering properties incorporated in the current CR-SIM were created using the T-matrix method based on assumptions regarding ice particle

composition and shape. More single-scattering properties from other scattering calculation methods can be incorporated by adding LUTs. Moreover, the gaseous attenuation will be considered in the future, as gaseous attenuation can be significant in the millimeter-wavelength radar measurements, and elevation angles will be corrected for the Earth's curvature. The analyses presented here serve as a reference to the CR-SIM package and illustrate numerous applications related to sampling uncertainty, sampling optimization, retrieval uncertainty, and comparison between models and observations.

**Appendix**

The radar observables are computed on the basis of the following equations. In the equations below, $M$ is the number of different hydrometeor species coexisting in the same spatial resolution volume of the model, and the subscript $i$ is index of specie, backward and forward scattering amplitudes are denoted as $f_{a,b}^{(\pi)}$ and $f_{a,b}^{(0)}$, respectively. The subscript * in an expression $[...]^*$ denotes conjugation, $R_e[...]$ and $I_m[...]$ represent the real and imaginary parts of the complex number, 510 respectively, and $|...|$ refers to the magnitude of the value between the single bars. $\lambda$ is the radar wavelength in millimeters, and $|Kw|^2$ is the dielectric factor (the value for water $= 0.92$ is used for all hydrometeor species in CR-SIM). The scattering amplitudes are given in millimeters. $N_i(D)$ defines the particle size distribution in terms of the number of particles per unit volume of air and unit bin size, given here in $m^{-3}\,mm^{-1}$, with the bin equivolume diameter $D$ in $mm$. Both the bin fall velocity, $V_{Fi}$, and vertical air velocity, $w$, are given in meters per second. The elevation and azimuth angles are denoted $\theta$ and $\varphi$ 515 respectively, and $u$ and $v$ are the two components of horizontal wind. The coefficients $A_{1i}$-$A_{7i}$ are the angular moments for one of the three horizontal orientation expressions taken from Ryzhkov et al. (2011).

$$Z_{hh} = \frac{4\,\lambda^4}{\pi^4\,|K_w|^2} \sum_{i=1}^{M} \left[ \int_0^\infty \left\{ \left|f_{bi}^{(\pi)}\right|^2 - 2R_e\left[f_{bi}^{(\pi)^*}\left(f_{bi}^{(\pi)} - f_{ai}^{(\pi)}\right)\right]A_{2i} + \left|f_{bi}^{(\pi)} - f_{ai}^{\pi}\right|^2 A_{4i} \right\} N_i(D)\,dD \right] \left[\frac{mm^6}{m^3}\right]$$

$$Z_{vv} = \frac{4\,\lambda^4}{\pi^4\,|K_w|^2} \sum_{i=1}^{M} \left[ \int_0^\infty \left\{ \left|f_{bi}^{(\pi)}\right|^2 - 2R_e\left[f_{bi}^{(\pi)^*}\left(f_{bi}^{(\pi)} - f_{ai}^{(\pi)}\right)\right]A_{1i} + \left|f_{bi}^{(\pi)} - f_{ai}^{\pi}\right|^2 A_{3i} \right\} N_i(D)\,dD \right] \left[\frac{mm^6}{m^3}\right]$$

$$Z_{vh} = \frac{4\,\lambda^4}{\pi^4\,|K_w|^2} \sum_{i=1}^{M} \left[ \int_0^\infty \left\{ \left|f_{bi}^{(\pi)} - f_{ai}^{\pi}\right|^2 A_{5i} \right\} N_i(D)\,dD \right] \left[\frac{mm^6}{m^3}\right]$$

$$Z_{DR} = \frac{Z_{hh}}{Z_{vv}} \quad [-]$$

$$LDR_h = \frac{Z_{vh}}{Z_{hh}} \quad [-]$$

$$K_{DP} = \frac{180\,\lambda}{\pi}\,10^{-3} \sum_{i=1}^{M} \left[ \int_0^\infty \left\{ R_e\left(f_{bi}^{(0)} - f_{ai}^0\right)A_{7i} \right\} N_i(D)\,dD \right] \left[\frac{^o}{km}\right]$$

$$\delta = \frac{180\,\lambda}{\pi}\ arg\left[\sum_{i=1}^{M}\left[\int_{0}^{\infty}\left\{\left|f_{bi}^{(\pi)}\right|^{2}\ +\ \left[\left|f_{bi}^{(\pi)}-f_{ai}^{(\pi)}\right|^{2}\right]A_{3i}\ -\ \left[f_{bi}^{(\pi)^{*}}\left(f_{bi}^{(\pi)}-f_{ai}^{(\pi)}\right)\right]A_{1i}\right.\right.\right.$$

$$\left.\left.\left.-\ \left[f_{bi}^{(\pi)}\left(f_{bi}^{(\pi)^{*}}-f_{ai}^{(\pi)^{*}}\right)\right]A_{2i}\right\}\ N_{i}\,(D)\,dD\right]\right]\ [°]$$

$$A_{h}=\frac{10}{log(10)}\ 2\,\lambda\ 10^{-3}\sum_{i=1}^{M}\left[\int_{0}^{\infty}\left\{\ I_{m}\big(f_{bi}^{(0)}\big)\ -\ I_{m}\big(f_{bi}^{(0)}-f_{ai}^{(0)}\big)\,A_{2i}\right\}\ N_{i}\,(D)\,dD\right]\ \left[\frac{dB}{km}\right]$$

$$A_{v}=\frac{10}{log(10)}\ 2\,\lambda\ 10^{-3}\sum_{i=1}^{M}\left[\int_{0}^{\infty}\left\{\ I_{m}\big(f_{bi}^{(0)}\big)\ -\ I_{m}\big(f_{bi}^{(0)}-f_{ai}^{(0)}\big)\,A_{1i}\right\}\ N_{i}\,(D)\,dD\right]\ \left[\frac{dB}{km}\right]$$

$$A_{DP}=\ A_{h}-\ A_{v}\ \left[\frac{dB}{km}\right]$$

$$VZ_{i}=\int_{0}^{\infty}\left\{\left|f_{bi}^{(\pi)}\right|^{2}\ -\ 2R_{e}\left[f_{bi}^{(\pi)^{*}}\left(f_{bi}^{(\pi)}-f_{ai}^{(\pi)}\right)\right]A_{2i}\ +\ \left|f_{bi}^{(\pi)}-f_{ai}^{\pi}\right|^{2}A_{4i}\right\}\ V_{fi}(D)\ N_{i}\,(D)\,dD\ \left[\frac{mm^{6}}{m^{3}}\frac{m}{s}\right]$$

$$V_{RW}=\frac{\frac{4\,\lambda^{4}}{\pi^{4}\,|K_{w}|^{2}}\ \sum_{i=1}^{M}[VZ_{i}\,]}{Z_{hh}}\ \left[\frac{m}{s}\right]$$

$$V_{D\_90}=\frac{\frac{4\,\lambda^{4}}{\pi^{4}\,|K_{w}|^{2}}\ \sum_{i=1}^{M}[w\,Z_{hh}\ -\ VZ_{i}\,]}{Z_{hh}}\ \left[\frac{m}{s}\right]$$

$$V_{D}=[u\,cos\varphi\ +\ v\,sin\varphi]\cos\theta\ +\ V_{D\_90}\sin\theta\ \left[\frac{m}{s}\right]$$

$SW_{H\_90}$

$$=\sqrt{\frac{\frac{4\,\lambda^{4}}{\pi^{4}\,|K_{w}|^{2}}\ \sum_{i=1}^{M}\left[\int_{0}^{\infty}\left\{\left|f_{bi}^{(\pi)}\right|^{2}\ -\ 2R_{e}\left[f_{bi}^{(\pi)^{*}}\left(f_{bi}^{(\pi)}-f_{ai}^{(\pi)}\right)\right]A_{2i}\ +\ \left|f_{bi}^{(\pi)}-f_{ai}^{\pi}\right|^{2}A_{4i}\right\}\ \left(V_{Fi}(D)-\frac{VZ_{i}}{Z_{hh}}\right)^{2}\ N_{i}\,(D)\,dD\right]}{Z_{hh}}}\ \left[\frac{m}{s}\right]$$

$$SW_{TOT}=\sqrt{SW_{H}^{2}\ +\ SW_{T}^{2}+\ SW_{S}^{2}+\ SW_{V}^{2}}\ \left[\frac{m}{s}\right]$$

where $SW_{H}$, $SW_{T}$, $SW_{S}$, and $SW_{V}$ are contributions from different hydrometeor terminal velocity, turbulence, mean wind shear, and cross wind. Tatarevic et al. (2019) describes detailed computations of these contributions.

The lidar observables are calculated as follows. For spherical droplets, using the BHMIE Mie code (Bohren and Huffman, 1998) the single particle extinction $\sigma_{\alpha}$ and backscattering cross sections $\sigma_{\beta}$ are computed for a lidar wavelength (i.e., 905, 532, and 353 $nm$).

$$\beta_{true}=\sum_{i}\sigma_{\beta}\,(D_{i})\ N\,(D_{i})\,\Delta D_{i}$$

$$\alpha_{ext}=\sum_{i}\sigma_{\alpha}\,(D_{i})\ N\,(D_{i})\,\Delta D_{i}$$

The value of refractive index of water used in the calculations is $1.327+i\,0.672\text{x}10^6$ (Hale and Querry, 1973). The attenuated backscatter, $\beta_{atten}$ at a distance $z$ can be written as:

$$\beta_{atten}(z) = \int_0^z \beta_{true}(z)\,\exp(-2\,\alpha_{ext}(z))\,dz,\;[sr\,m]^{-1}$$

where $\beta_{true}$ is the true backscatter at height z, and $\alpha_{ext}$ is the extinction coefficient:

$$\beta_{true} = \frac{1}{4\pi}\sum_i \sigma_\beta(D_i)\,N(D_i)\,\Delta D_i\quad[sr\,m]^{-1}$$

$$\alpha_{ext} = \frac{1}{4\pi}\sum_i \sigma_\alpha(D_i)\,N(D_i)\,\Delta D_i\quad[sr\,m]^{-1}$$

**List of acronyms**

| | |
|---|---|
| ARM | Atmospheric Radiation Measurement Facility |
| ARSCL | Active Remotely-Sensed Cloud Location |
| CDF | Cumulative distribution function |
| CFP | Cloud fraction profile |
| CRM | Cloud resolving model |
| CR-SIM | Cloud resolving model Radar SIMulator |
| C-SAPR | C-band scanning ARM precipitation radar |
| CWRHI | Cross-wind range-height indicator |
| GCM | Global climate model |
| GNU GPL | GNU General Public License |
| KAZR | Ka-band ARM Zenith-pointing Radar |
| LASSO | LES ARM Symbiotic Simulation and Observation |
| LES | Large eddy simulation |
| LUT | Look-up table |
| MAPE | Mean absolute percentage error |
| MC3E | Midlatitude Continental Convective Clouds Experiment |
| MCS | Mesoscale convective system |
| MF | Mass flux |
| MPL | Micro pulse lidar |
| OSSE | Observing system simulation experiment |
| PPI | Plan position indicator |
| PSD | Particle size distribution |
| RHI | Range-height indicator |





| | RMSE | Root mean square error |
|---|---|---|
| 575 | SCR | Scanning cloud radar |
| | SGP | Southern Great Plains |
| | UF | Updraft fraction |
| | VCP | Volume coverage pattern |
| | WRF | Weather Research Forecasting model |
| 580 | X-SAPR | X-band scanning ARM precipitation radar |

**Code and data availability**.

The source code for CR-SIM, along with downloading, installation instructions, and user guide is available at the Stony Brook

University Academic Commons (https://commons.library.stonybrook.edu/somasdata/4), https://www.bnl.gov/CMAS/cr-sim.php (last access: September 6, 2019), and https://you.stonybrook.edu/radar/research/radar-simulators/ (last access: September 5, 2019). The software is licensed under GNU General Public License. A code that converts model grid coordinates to radar polar coordinates is also available at the Stony Brook University Academic Commons https://commons.library.stonybrook.edu/somasdata/4 and https://you.stonybrook.edu/radar/research/radar-simulators/ (last

access: August 30, 2019). There is ongoing work to integrate this module into the CR-SIM package. The CR-SIM package available online includes a configuration file and a script to run the code. The LASSO data used in the manuscript are available at the ARM archive: https://adc.arm.gov/lassobrowser (ARM, 2017). All configuration files used in the simulations and other input data available online https://commons.library.stonybrook.edu/somasdata/3 (Oue et al. 2019b).

**Author contributions**.

M. Oue and P. Kollias designed the OSSE experiments, and M. Oue carried them out. A. Tatarevic developed the radar simulator code, and with M. Oue, D. Wang, and K. Yu contributed to evolve, improve, and optimize the code. M. Oue prepared the manuscript with contributions from all co-authors.

**Competing interests**.

The authors declare that they have no conflict of interest.

**Acknowledgements.**

We would like to thank H. Morrison, Z. Feng, J. Fan, T. Matsui, and S. Endo for providing the WRF output data and for their valuable suggestions. We extend our gratitude to M. Mech and A. Hansen, P. Marinescu, and T. Yamaguchi for providing the ICON, RAMS, and SAM output data, respectively, and for their helpful comments. We would also like to thank M. I.

Mishchenko for making his T-matrix codes public and freely available for research purposes and J. Vivekanandan for his Mueller-matrix-based code, which was used to validate the computation of radar variables in CR-SIM. This research was supported by the Climate Model Development and Validation activity funded by the Office of Biological and Environmental Research in the US Department of Energy Office of Science through award KP170304 (P. Kollias and A. M. Vogelmann). D. Wang of Brookhaven Science Associates, LLC, is supported under Contract DE-SC0012704 with the US DOE.

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

**Tables**

Table 1. The minimum and maximum sizes and bin spacing of simulated particles for each hydrometeor category from the bulk microphysics schemes except the P3 scheme. 'Particle size' here refers to the particle maximum dimension.

| Category | Minimum size [$\mu$m] | Maximum size [$\mu$m] | Bin spacing [$\mu$m] |
|---|---|---|---|
| Cloud | 1 | 250 | 1 |
| Drizzle | 1 | 250 | 1 |
| Rain | 100 | 9000 | 20 |
| Ice | 1 | 1496 | 5 |
| Snow, aggregates | 100 | 50000 | 100 |
| Graupel | 5 | 50005 | 100 |
| Hail | 5 | 50005 | 100 |

Table 2. Scattering types built in the current LUTs and settings for the scattering calculations. Diameter ranges are the same as shown in Table. 1.

| Hydrometeor type | Scattering type | Temperature [°C] | Density [g cm$^{-3}$] | Shape |
|---|---|---|---|---|
| *cloud* | cloud | -30 to 20 every 2 | 1 | Spherical |
| *rain* | raina | 0–20 every 2 | 1 | Oblate according to Andsager et al. (1999) |
| | rainb | | | Oblate according to Brandes et al. (2002) |
| *ice* | ice_ar0.20 | -30 | 0.4–0.9 every 0.1 | Oblate with aspect ratio fixed to 0.2 |
| | ice_ar0.90 | | | Oblate with aspect ratio fixed to 0.9 |
| | Smallice (P3 only) | | 0.001 and 0.9 | Spherical |
| *snow* | snow_ar0.60 | -30 | 0.01, 0.05, and 0.1–0.5 every 0.1 | Oblate with aspect ratio fixed to 0.6 |
| *graupel hail* | graupel (P3 graupel only) | -30 | 0.01–0.09 every 0.01 and 0.1–0.6 every 0.05 | Spherical |
| | graupel_ar0.80 | | 0.4, 0.5, and 0.9 | Oblate with aspect ratio fixed to 0.8 |
| | gh_ryzh | | | Oblate according to Ryzhkov et al. (2011) |
| *unrimed ice (P3 only)* | unrimedice_ar0.40 | -30 | 0.001 and 0.005 | Oblate with aspect ratio fixed to 0.4 |
| | unrimedice_ar0.60 | | 0.01–0.09 every 0.1 | Oblate with aspect ratio fixed to 0.6 |
| | unrimedice_ar0.80 | | | Oblate with aspect ratio fixed to 0.8 |
| | Unrimedice | | 0.1–0.8 every 0.05 | Spherical |
| *partially rimed ice (P3 only)* | partrimedice_ar0.40 | -30 | 0.01–0.09 every 0.01 | Oblate with aspect ratio fixed to 0.4 |
| | partrimedice_ar0.60 | | | Oblate with aspect ratio fixed to 0.6 |
| | partrimedice_ar0.80 | | | Oblate with aspect ratio fixed to 0.8 |
| | Partrimedice | | 0.1–0.6 every 0.05 | Spherical |

Table 3. Incorporated microphysics schemes and corresponding CRMs.


| CRM | Microphysics scheme (M=moment) |
|---|---|
| Weather Research and Forecasting Model (WRF) | Morrison 2-M scheme (Morrison et al. 2005) |
| | Milbrandt and Yau multi-M scheme (Milbrandt and Yau 2005a, 2005b) |
| | Thompson 1- and 2-M scheme (Thompson et al. 2008) |
| | Predicted particle properties (P3) scheme (Morrison and Milbrandt 2015) |
| | Spectral bin microphysics (Fan et al. 2012) |
| ICOsahedral Non-hydrostatic general circulation model (ICON) | Seifert and Beheng 2-M scheme (Seifert and Beheng 2006; Seifert 2008) |
| Regional Atmospheric Modeling System (RAMS) | 2-M scheme (Cotton et al., 2003) |
| System for Atmospheric Modeling (SAM) | Tel Aviv University 2-M bin microphysics (Tzivion et al. 1987; Feingold et al. 1996) |
| | Morrison 2-M scheme (Morrison et al. 2005) |

Table 4. Computed radar variables. Units the variables stored in output files are provided in square brackets.

| Variable | Description |
|---|---|
| $Z_{hh}$ | Radar reflectivity factor at horizontal polarization [dBZ] |
| $Z_{vv}$ | Radar reflectivity factor at vertical polarization [dBZ] |
| $Z_{vh}$ | Cross-polarization radar reflectivity factor [dBZ] |
| $Z_{DR}$ | Differential reflectivity, defined as the ratio between the fraction of horizontally polarized backscattering and vertically polarized backscattering [dB] |
| $LDR_h$ | Linear depolarization ratio, defined as the ratio of the power backscattered at vertical polarization to the power backscattered at horizontal polarization for a horizontally polarized field [dB] |
| $K_{DP}$ | Specific differential phase, the backward propagation phase difference between the horizontally and vertically polarized waves at a specific distance [° km$^{-1}$] |
| $\delta$ | Differential backscatter phase, defined as the difference between the phases of horizontally and vertical polarized components of the wave caused by backscattering from the objects in the radar resolution volume, computed based on Trömel et al (2013) [°] |
| $A_h$ | Specific attenuation at horizontal polarization, or for horizontally polarized waves, represented by forward scattering amplitudes [dB km$^{-1}$] |
| $A_v$ | Specific attenuation at vertical polarization, or for vertically polarized waves, represented by forward scattering amplitudes [dB km$^{-1}$] |
| $A_{DP}$ | Specific differential attenuation, defined as the difference between the specific attenuations for horizontally and vertically polarized waves [dB km$^{-1}$] |
| $V_D$ | Mean radial Doppler velocity (positive away from the radar) [m s$^{-1}$] |
| $V_{D\_90}$ | Mean vertical Doppler velocity (positive upward) [m s$^{-1}$] |
| $SW_{TOT}$ | Spectrum width, including contribution of four major spectral broadening mechanisms (Doviak and Zrnić, 2006): 1) different hydrometeor terminal velocity of different sizes $SW_H$, 2) turbulence, 3) mean wind shear contribution, and 4) cross wind contribution. Antenna motion and contributions due to variation of orientation and vibrations of hydrometeor are not considered. [m s$^{-1}$] |
| $SW_{H\_90}$ | Spectrum width due to different hydrometeor terminal velocity of different sizes in vertical, such that $SW_{H\_90} = SW_H$ ($\theta$=90°), where $\theta$ is the elevation angle measured from horizontal [m s$^{-1}$] |
| $V_{RW}$ | Reflectivity weighted velocity (positive downward) [m s$^{-1}$] |
| $Z_{MIN}$ | Radar minimum detectable reflectivity [dBZ] |
| Spectra_$Z_{hh}$ | Radar Doppler spectra at horizontal polarization [m s$^{-1}$ dB$^{-1}$] |
| Spectra_$Z_{vv}$ | Radar Doppler spectra at vertical polarization [m s$^{-1}$ dB$^{-1}$] |
| Spectra_$Z_{vh}$ | Cross-polarization radar Doppler spectra [m s$^{-1}$ dB$^{-1}$] |

Table 5. Computed lidar variables

| Variable | Description |
|---|---|
| $\beta_{hydro}$, $\beta_{aero}$, $\beta_{mol}$ | Backscatter [sr$^{-1}$ m$^{-1}$] for cloud droplets and cloud ice ($\beta_{hydro}$), aerosols ($\beta_{aero}$), and air molecules ($\beta_{mol}$) |
| $\beta_{hydro\_atten}$, $\beta_{aero\_atten}$, $\beta_{mol\_atten}$ | Attenuated backscatter [sr$^{-1}$ m$^{-1}$] for cloud droplets and cloud ice ($\beta_{hydro\_atten}$), aerosols ($\beta_{aero\_atten}$), and *air* molecules ($\beta_{mol\_atten}$) |
| $\alpha_{ext\_hydro}$, $\alpha_{ext\_aero}$ | Extinction coefficient [m$^{-1}$] for cloud droplets and cloud ice ($\alpha_{ext\_hydro}$) and aerosols ($\alpha_{ext\_aero}$) |
| $\beta_{total}$ | Total backscatter [sr$^{-1}$ m$^{-1}$], defined as $\beta_{total} = \beta_{hydro} + \beta_{aero} + \beta_{mol}$ |
| $\beta_{total\_atten}$ | Attenuated total backscatter [sr$^{-1}$ m$^{-1}$], defined as $\beta_{total\_atten} = \beta_{hydro\_atten} + \beta_{aero\_atten} + \beta_{mol\_atten}$ |
| $S$ | Lidar ratio, defined as $S = \alpha_{ext\_hydro} / \beta_{hydro}$ |

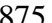

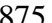

**Figure 1: Radar observables produced by CR-SIM for a mesoscale convective system. The system was simulated using WRF with the Morrison 2-moment microphysics scheme at 1.8 km altitude. Shown are horizontal cross sections of (a) total hydrometeor content and (b) vertical air velocity from the WRF simulation. CR-SIM produces the following parameters for a scanning S-band (3 GHz) radar located at the center of the domain: (c) $Z_{hh}$, (d) $Z_{DR}$, (e) $K_{DP}$, (f) radar antenna elevation angle, (g) Doppler velocity, and (h) spectrum width at 12:18:00 UTC.**


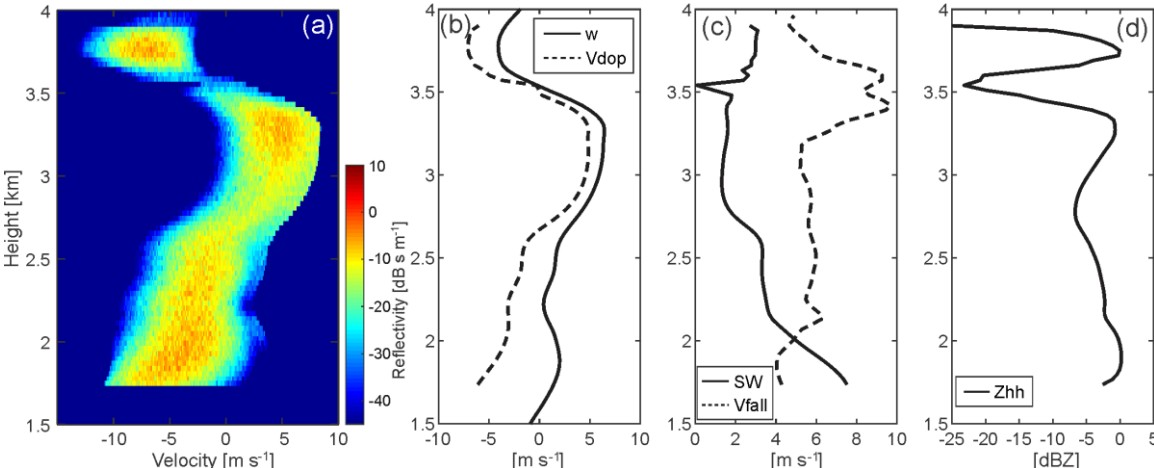

**Figure 2: CR-SIM examples of radar observables for a shallow convection LASSO case from a WRF simulation coupled with the Thompson microphysics scheme. Shown are (a) simulated radar Doppler spectra, (b) model vertical velocity (w, solid line) and simulated mean Doppler velocity (Vdop, dashed line), (c) simulated spectrum width (SW, solid line) and simulated reflectivity-weighted velocity (Vfall, dashed line), and (d) simulated total reflectivity ($Z_{hh}$) at S band (3 GHz). In (a) and (b), a positive sign indicates upward motion, and in (c), a positive sign indicates downward motion (fall speed).**



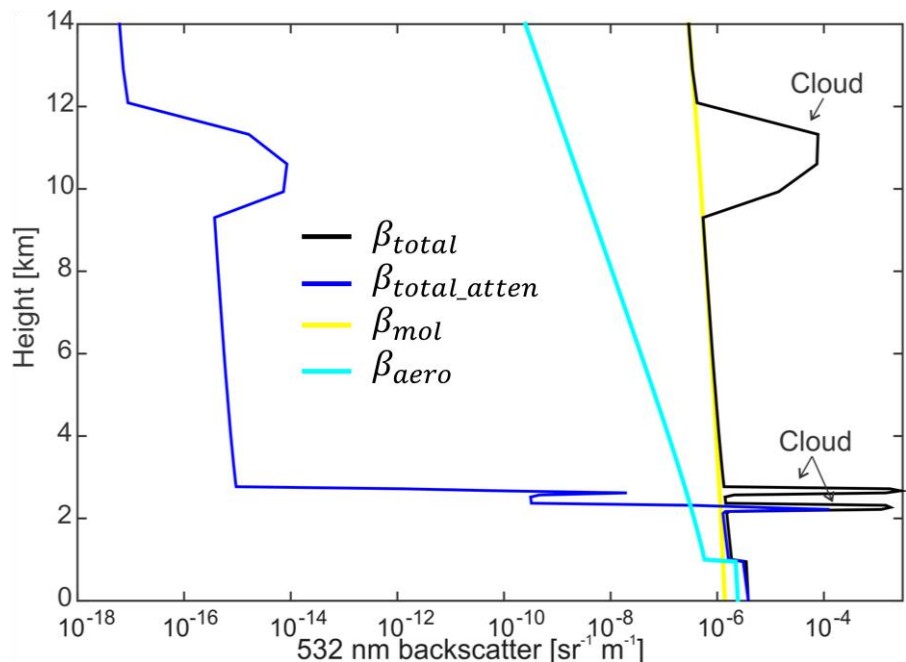

**Figure 3: Lidar observables from CR-SIM for a cumulus case from LASSO using WRF with the Morrison 2-moment microphysics scheme. Example of simulated vertical profiles are shown for $\beta_{total}$, $\beta_{total\_atten}$, $\beta_{mol}$, and $\beta_{aero}$ at a wavelength of 532 *nm*.**


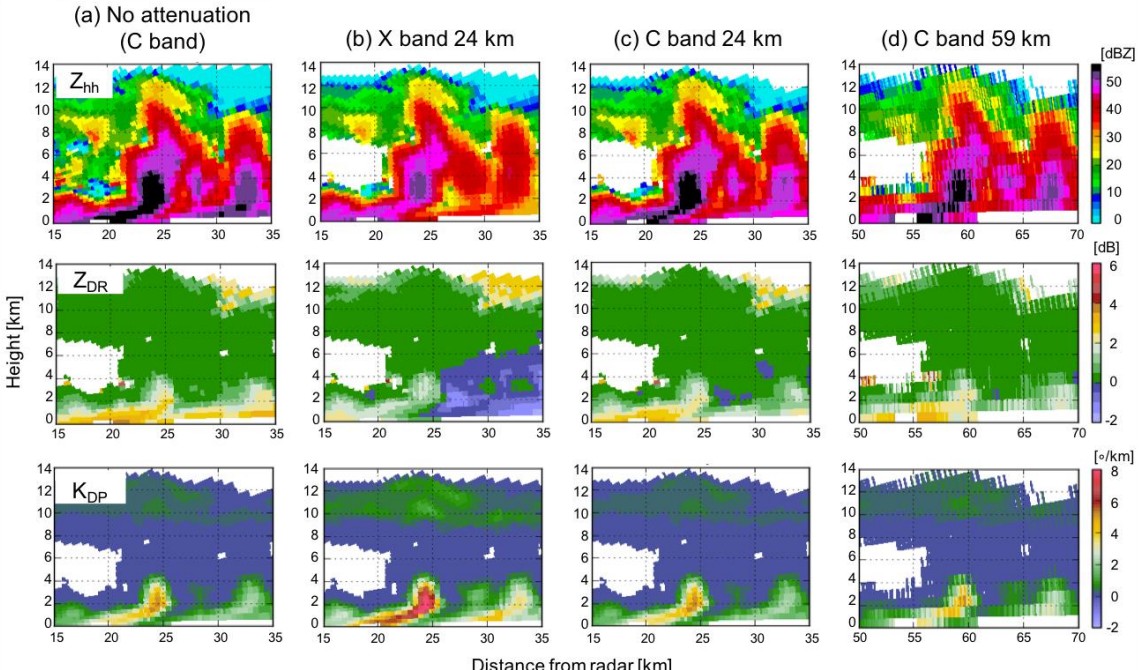

**Figure 4: Examples of C- and X-band (5.5 and 9 GHz, respectively) RHI scans with a beamwidth of 1° produced using CR-SIM for a convective cell in a mesoscale convective system (MCS). The simulation uses WRF with the Morrison double moment microphysics scheme for an MCS on May 20, 2011 during MC3E. Shown are variables at X- and C-band frequencies 15-35 km from the radar as a function of height at 12:18:00 UTC: (top raw) $Z_{hh}$, (middle raw) $Z_{DR}$, and (bottom raw) $K_{DP}$. The figure shows (a) C-band variables without attenuation, (b) X-band variables with attenuation from a radar 24 km from the convective cell, (c) C-band variables with attenuation from a radar 24 km away, and (d) C-band variables from a radar located 59 km away with attenuation.**

**Input**
CRM/LES data
- Environmental parameters (P, T, $\rho_{dry}$, H, $Q_{vapor}$)
- Hydrometeor parameters ($Q_i$, $N_i$, $\rho_i$)
- Dynamical parameters (u, v, w)

**Output 1**
Idealized observables at each model grid box

**Forward Model**
*Radar and lidar observables for each hydrometeor type accounting for PSD using T-matrix*

**Output 2**
Virtual observables including instrument characteristics and limitations

**Instrument Model**
- *Sampling geometry*
- *Sampling volume (beam width, scan strategy)*
- *Attenuation*
- *Sensitivity*

**Output 3**
Virtual value added products (e.g., u, v, w, cloud location, Q, N)

**Retrieval Model**
*Algorithms retrieving cloud and precipitation properties from measurements*

**Figure 5: Diagram for CR-SIM and its applications. The diagram indicates the CR-SIM input and the different levels of output for the forward model, instrument model, and retrieval model.**


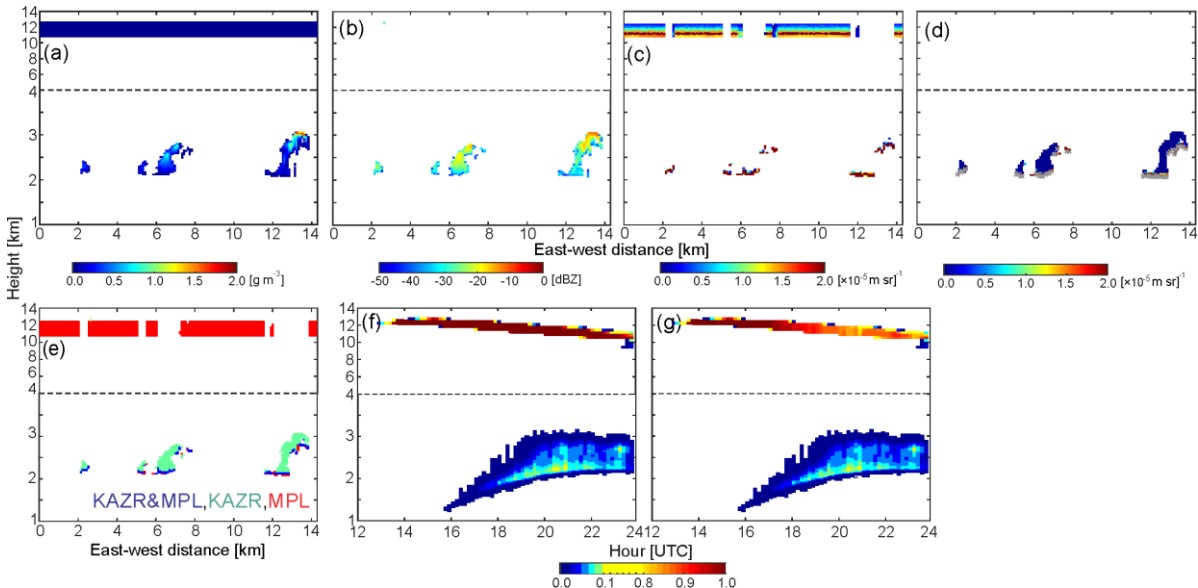


**Figure 6: Simulated vertically pointing radar and lidar measurements and the ARSCL product for a shallow convection case on June 27, 2015. (a-e) Vertical cross sections of (a) water content from the WRF model, (b) Ka-band (35 GHz) radar reflectivity accounting for radar sensitivity and attenuation, (c) MPL attenuated backscatter, (d) ceilometer backscatter (colorbar) and first cloud base (gray dots), and (e) the ARSCL cloud mask. (f,g) Height-versus-time cross** 925 **sections of domain-averaged cloud fraction from (f) WRF water content > 0.001 g m$^{-3}$ and (g) the simulated ARSCL product.**

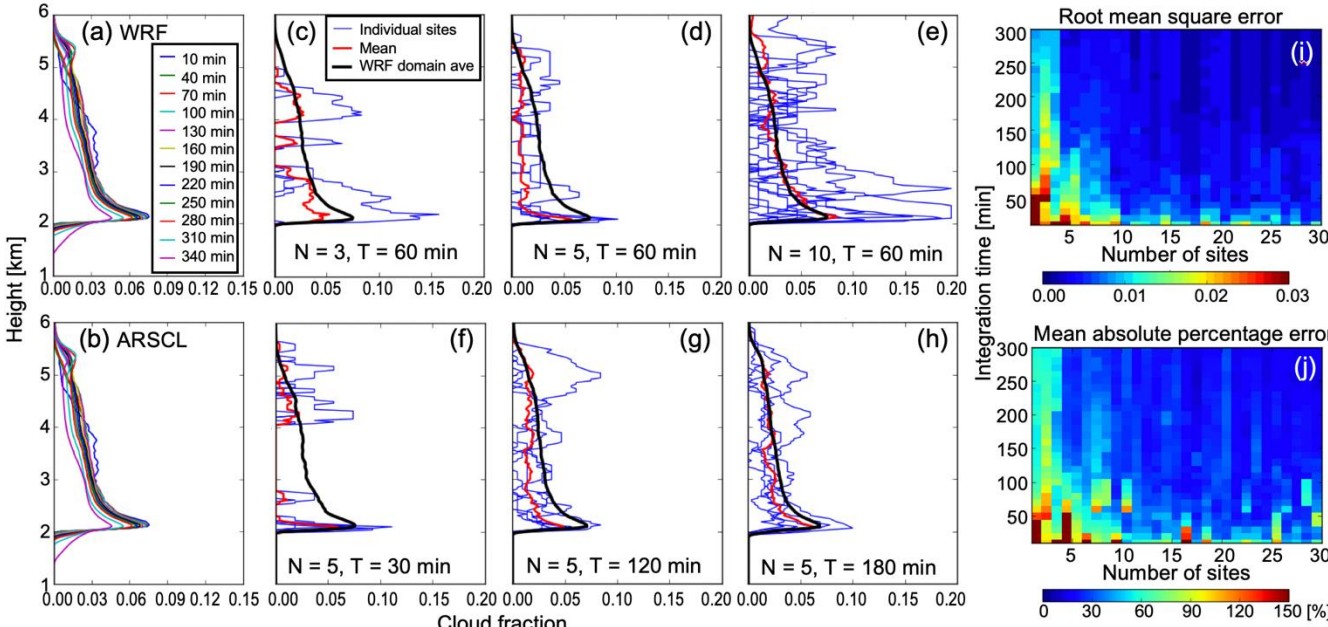


**Figure 7: Investigation of errors of cloud fraction profiles (CFPs) from profiling measurements. CFPs from single sites are estimated by integrating over time and then averaged. Shown are domain-averaged cloud fraction profiles (CFPs) from (a) WRF-simulated cloud water mixing ratio and (b) the simulated ARSCL product for a shallow convection case**
**on June 11, 2016. Colors in (a) and (b) represent different integration time periods centered at 21:00:00 UTC. The minimum threshold for the WRF cloud water mixing ratio is 0.01 g kg$^{-1}$. (c-h) CFPs from the simulated ARSCL with different number of observation sites N and different integration periods T. The black line in (c-h) represents the domain-averaged CFP from the WRF-simulated cloud water mixing ratio, blue lines represent CFPs from individual observation sites, and the red line represents the mean CFP from averaging over the individual sites. (i and j) Root**
**mean square error (i) and mean absolute percentage error (j) of the simulated ARSCL CFPs as a function of the number of observation sites and integration period.**

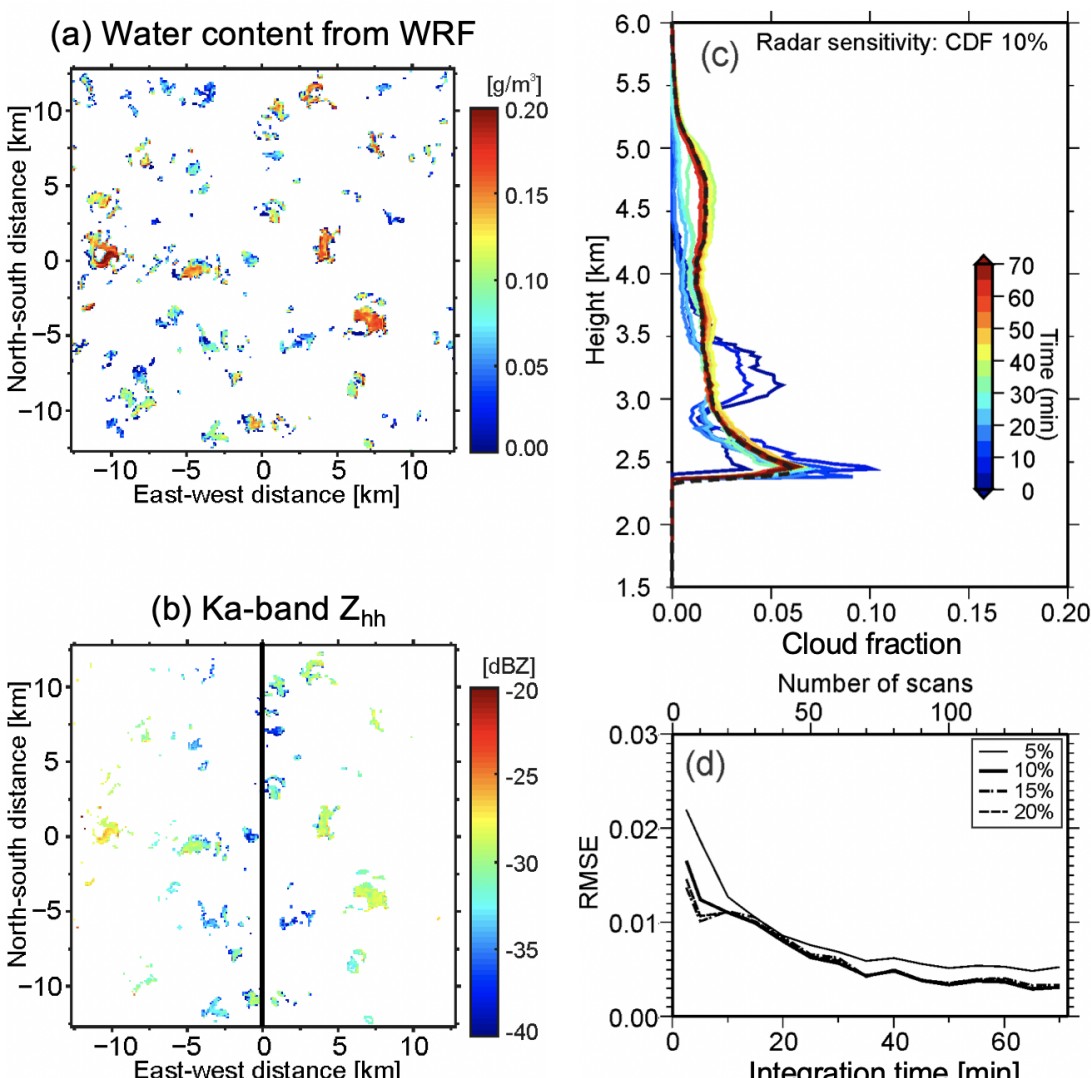

**Figure 8:** Horizontal cross sections of (a) water content simulated by WRF and (b) Ka-band (35 GHz) $Z_{hh}$ simulated at
2.4 km above ground level for a LASSO case. In (b), it is assumed that the radar is located at x=0 km and the RHI is
scanned along the east-west axis, and the radar sensitivity $Z_{MIN}$ with $Z_0$ =-50 dBZ was applied. (c) Cloud fraction profiles
corresponding to the 10% cumulative distribution function (CDF) isoline with changing integration time of CWRHI
(hence, number of scans). (d) The root-mean-square error (RMSE) from the LES domain-averaged CFP for CDF
isolines of 5% (thin solid line), 10% (thick solid line), 15% (dashed line), and 20% (dashed-dotted line) as a function of
integration time. The black dashed line in (c) represents the LES domain-averaged CFP for hydrometeor mixing ratio
$\geq 0.01$ g kg$^{-1}$. (c) and (d) are adapted from Oue et al. (2016).

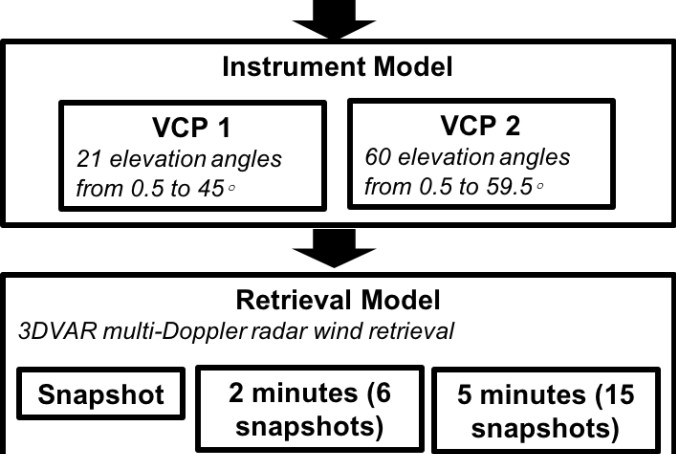


**Figure 9:  A diagram of an Observing System Simulation Experiment study to investigate the impacts of radar volume coverage pattern (VCP) on a multi-Doppler radar wind retrieval.**





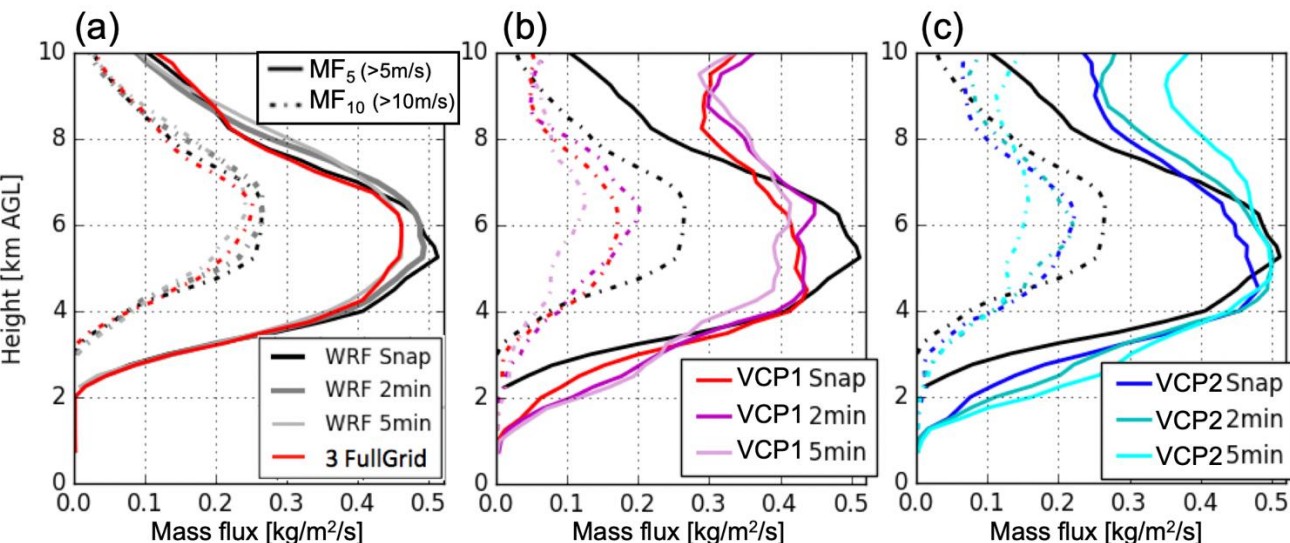

**Figure10: Vertical profiles of convective mass flux with updraft thresholds of 5 m s$^{-1}$ (solid lines) and 10 m s$^{-1}$ (dashed lines). Displayed in each panel are different retrieval simulations represented by the colors. The dark gray line in (a) represents the time average of the WRF output over 2 minutes, and the light gray line in (a) represents the time average of the WRF output over 5 minutes. The profile from the WRF snapshot is displayed in each panel by a black solid line. Adapted from Oue et al. (2019a).**

