# Peer review of "The Cloud Resolving Model Radar Simulator (CR-SIM) Version 3.3: Description and Applications of a Virtual Observatory"

_Geoscientific Model Development, 2019_

## Referee Comment (RC1) · Anonymous Referee #1 · 17 Oct 2019

The manuscript describes CR-SIM, a radar and multi-instrument emulator that has a wide range of potential applications. CR-SIM is freely available to the scientific community and has the interfaces necessary to be broadly used (e.g., compatibility with major community models, physics schemes and data formats). Thus, this is an impactful study and provides an important overview of the tool and its applications.

The manuscript layout is excellent and easy to follow. Overall, the methods employed are sound and the analyses are well described. The only major comment I have is that the description of the radar emulator needs more specific details as described below. I also have some other minor and technical comments to consider.

[Figure]

General comments: 1) Since the goal of this journal is to make the model reproducible, more equations, references, and is needed for the following items to achieve this:

a. How are the model PSDs transformed into radar variables? The authors acknowledge Dr. Vivekanandan's the Mueller Matrix code at the end, but I didn't see this cited or described in the text.

b. How is the radar antenna pattern and pulse emulated? What are the range and antenna weighting patterns and are there different options?

c. How are mixed phase hydrometeors treated in the calculation of scattering amplitudes, and how are the mixing ratios of these particles computed from pure liquid and pure ice hydrometeor classes in the models?

2) To give readers a sense of the computational requirements and burden for running a simulation, can you please describe what computing platforms were used for these simulations and what the simulation run times are?

Specific Comments:

Lines 73 – 87: A little deeper treatment of past radar simulators and where the authors' contribution fits is warranted. For example, aside from the applications, this will enable the reader to more clearly see what the strengths and weaknesses of the radar emulator are and how they compare to other emulator tools. For example, some radar emulators such as Snyder et al. (2017a,b) apply a radar forward simulator to the model grid cells whereas other simulators account for the radar observing geometry (e.g., beamwidth, range resolution). Other simulators emulate radar time series signals based on model turbulence whereas others do not. Finally, some simulators take into account complex electromagnetics of hydrometeors or other weather radar observed scatterers.

C. Capsoni, M. D'Amico, and R. Nebuloni, 2001: A multiparameter polarimetric radar simulator. J. Atmos. Ocean. Technol., 18, 1799–1809.

Caumont, O., V. Ducrocq, G. Delrieu, M. Gosset, J. Pinty, J. Parent du Châtelet, H. Andrieu, Y. Lemaître, and G. Scialom, 2006: A Radar Simulator for High-Resolution Nonhydrostatic Models. J. Atmos. Oceanic Technol., 23, 1049–1067, https://doi.org/10.1175/JTECH1905.1.

B. L. Cheong, R. D. Palmer, and M. Xue, 2008: A time series weather radar simulator based on high-resolution atmospheric models. J. Atmos. Ocean. Technol., 25, 230–243.

Jiang, Z., M.R. Kumjian, R.S. Schrom, I. Giammanco, T. Brown-Giammanco, H. Estes, R. Maiden, and A.J. Heymsfield, 2019: Comparisons of Electromagnetic Scattering Properties of Real Hailstones and Spheroids. J. Appl. Meteor. Climatol., 58, 93–112, https://doi.org/10.1175/JAMC-D-17-0344.1.

Line 90: and spectrum width?

Line 105: What particle size spacing is used and what are the minimal and maximum sizes of particles simulated? The truncation can affect the resulting simulated measurements.

Lines 105 – 114: In general, I could follow the authors' description of the scattering properties and implement it into a simulator. However, there is no description of how mixed phase hydrometeors are treated. How is this accomplished?

Line 197: Suggest "for convective cells" since multiple convective cells are evident in the image

Line 251: Should add a reference for the Morrison microphysics scheme

Line 282: It isn't clear which simulation output is saved every 10 minutes (CR-SIM or WRF LES), or both.

Lines 289 – 290: Is spatial or temporal sampling driving these major errors?

Line 332: Should this say minimum detectable reflectivity Zmin, similar to a KASCR?

Can you please provide the sensitivity of the simulated radar?

Lines 342 – 343: How does this compare to the current CWRHI measurement interval?

Lines 351 – 352: References needed for multi-Doppler error sources

Line 356: Which advection-correction technique? Please state and cite

Lines 364 – 366: While this study is examining a hypothetical scenario for VCPs, is the 60 elevation angle scenario practical for the listed update intervals of 2 and 5 minutes? This would require PPI scans every 2 or 5 seconds which is not possible with the X-SAPR (but is with other X-band radar systems). Please elaborate on the technology limitation.

Figure 1: The Doppler velocity and spectrum with colors are saturated in a large portion of the figures. Suggest a wider colorbar range.

Techincal Corrections: Line 280: Suggest "highly heterogeneous" instead of "high heterogenous"

Line 283: Suggest "between 10-minute intervals"

Line 302: Suggest "Each panel shows that CFPs at a single site"

Line 695: Extra comma in the data "May, 20, 2011"

Figure 7 caption: units for cloud water mixing ratio should be g/kg

Figure 9: Suggest "20-second output for 5 minutes" to be more clear in the Forward Model box

[Figure]

---

## Author Comment (AC1) · 29 Oct 2019

**The manuscript describes CR-SIM, a radar and multi-instrument emulator that has a wide range of potential applications. CR-SIM is freely available to the scientific community and has the interfaces necessary to be broadly used (e.g., compatibility with major community models, physics schemes and data formats). Thus, this is an impactful study and provides an important overview of the tool and its applications.**
**The manuscript layout is excellent and easy to follow. Overall, the methods employed are sound and the analyses are well described. The only major comment I have is that the description of the radar emulator needs more specific details as described below. I also have some other minor and technical comments to consider.**

Thank you very much for your comments and suggestions. Providing this valuable feedback has helped to improve the current manuscript. We have modified the manuscript, taking into account the referee's comments. The following contains our detailed responses to referee's comments, with our responses in plain type given underneath your original comments in bold type.

**General comments:**
**1) Since the goal of this journal is to make the model reproducible, more equations, references, and is needed for the following items to achieve this:**
**a. How are the model PSDs transformed into radar variables? The authors acknowledge Dr. Vivekanandan's the Mueller Matrix code at the end, but I didn't see this cited or described in the text.**

The radar observables are computed by integrating scattering properties over the discrete PSD using a constant size bin for each hydrometeor. The complex scattering amplitudes for equally spaced particle sizes are pre-computed and stored in the look-up tables using the Mishchenko's T-matrix code for single non-spherical particles at a fixed orientation. Using the calculated scattering amplitudes, we computed radar observables following Ryzhkov et al. (2011), accounting for an assumption of the orientation distribution which can be selected by the user. We described the information in section 2 in the revised manuscript. Because the equations of radar observables have been well described in Ryzhkov et al. (2011) and are not unique to CR-SIM, we decided to not add them in the manuscript. The method is fully described in the Section 4.4 and 4.5 in the CR-SIM User Guide (ftp://ftp.radar.bnl.gov/outgoing/moue/crsim/docs/crsim-UserGuide-v3.3.0.pdf).

To verify our computations of radar variables, we employed Dr. Vivekanandan's Mueller-matrix-based code from Vivekanandan et al. (1991). The figure below shows an example of a comparison of radar reflectivity for raindrops computed using the methods in CR-SIM and the Mueller-matrix-based code. The results show consistent values. Figures 4 and 5 in the CR-SIM User Guide compare all radar variables at 3 GHz and 9.5 GHz for raindrops. We revised the acknowledgement section to state how the Mueller-matrix-based code was used.

[Figure]

Figure: Comparison of Radar reflectivity at 3 GHz for raindrops as a function of elevation and particle size (diameter of an equi-volume sphere) computed by (a) method used in the CR-SIM based on Mishchenko's T-matrix code for a non-spherical particle at a fixed orientation and Ryzhkov's formulas for angular moments, and (b) the Mueller-matrix-based code from Vivekanandan et al. (1991).

**b. How is the radar antenna pattern and pulse emulated? What are the range and antenna weighting patterns and are there different options?**

We did not emulate the radar antenna pattern in CR-SIM. In the post-processing instrument model to convert grid data into radar polar coordinate data, we simply use a Gaussian function as a radar directivity function and average simulated radar observables within a range-gate bin with the gaussian weighting in the beam angle and range directions. We added the information to Section 2.3.

**c. How are mixed phase hydrometeors treated in the calculation of scattering amplitudes, and how are the mixing ratios of these particles computed from pure liquid and pure ice hydrometeor classes in the models?**

CR-SIM treats hydrometer categories for which mixing ratio (and/or number density) were predicted in the input cloud model using the selected microphysics scheme. At this stage, all ice hydrometeors (e.g., snow, ice, graupel, hail) are modeled as dielectrically dry spheroids i.e., assuming the dry growth of larger ice particles. Thus, the refractive index of dielectrically dry hydrometeors depends on relative mixture of air and solid ice. In other words, the refractive index depends on hydrometeor density and is computed using Maxwell Garnett (1904) mixing formula.

**2) To give readers a sense of the computational requirements and burden for running a simulation, can you please describe what computing platforms were used for these simulations and what the simulation run times are?**

We used a computer having 500-GB memory and 24 processors (Intel(R) Xeon(R) CPU E5-2670 v3 @ 2.30GHz) with 12 cores each for the simulations presented in the manuscript. The runtime depends on how many threads are used, simulation domain size, and the numbers of cloudy gridboxes. The following is an example of computer resources and runtime for the simulation of the MCS case in Fig. 1.

Domain size: 667 x 667 x 12
Number of threads used: 16
Runtime: 270 sec

**Specific Comments:**
**Lines 73 – 87: A little deeper treatment of past radar simulators and where the authors' contribution fits is warranted. For example, aside from the applications, this will enable the reader to more clearly see what the strengths and weaknesses of the radar emulator are and how they compare to other emulator tools. For example, some radar emulators such as Snyder et al. (2017a,b) apply a radar forward simulator to the model grid cells whereas other simulators account for the radar observing geometry (e.g., beamwidth, range resolution). Other simulators emulate radar time series signals based on model turbulence whereas others do not. Finally, some simulators take into account complex electromagnetics of hydrometeors or other weather radar observed scatterers.**
**C. Capsoni, M. D'Amico, and R. Nebuloni, 2001: A multiparameter polarimetric radar simulator. J. Atmos. Ocean. Technol., 18, 1799–1809.**
**Caumont, O., V. Ducrocq, G. Delrieu, M. Gosset, J. Pinty, J. Parent du Châtelet, H. Andrieu, Y. Lemaître, and G. Scialom, 2006: A Radar Simulator for High-Resolution Nonhydrostatic Models. J. Atmos. Oceanic Technol., 23, 1049–1067, https://doi.org/10.1175/JTECH1905.1.**
**B. L. Cheong, R. D. Palmer, and M. Xue, 2008: A time series weather radar simulator based on high-resolution atmospheric models. J. Atmos. Ocean. Technol., 25, 230–243.**
**Jiang, Z., M.R. Kumjian, R.S. Schrom, I. Giammanco, T. Brown-Giammanco, H. Estes, R. Maiden, and A.J. Heymsfield, 2019: Comparisons of Electromagnetic Scattering Properties of Real Hailstones and Spheroids. J. Appl. Meteor. Climatol., 58, 93–112, https://doi.org/10.1175/JAMC-D-17-0344.1.**

One of the features of using CR-SIM is it produces both radar and lidar observables for all the cloud resolving model grid boxes accounting for elevation angles relative to a radar location, similar to Snyder et al. (2017a,b), rather than other radar simulators that account for radar geometry characteristics such as beamwidth and radar range resolution to simulate scatters within the radar resolution volume (e.g., Capsoni et al, 2001; Caumont et al., 2006; Cheong et al., 2008). The radar sampling characteristics are accounted for in the post processing instrument model, as explained in the response to the referee's comment #1b. This feature facilitates the process of configuring any desirable observational setup with a varying number of profiling or scanning sensors.

The LUTs of scattering properties incorporated in the current CR-SIM were created using the T-matrix method where solid phase hydrometeors are represented as dielectrically dry oblate spheroids. These assumptions are rather simple compared to some other radar simulators which take into account complex electromagnetic scattering by mixed-phase hydrometeors or ice hydrometeors with possibly irregular shapes (e.g., Snyder et al., 2017a,b; Jiang et al., 2019). However, such complex electromagnetic scatters can be easily incorporated by adding LUTs of their scattering properties from different scattering calculation methods. In CR-SIM, a "scattering type" refers to each hydrometeor class for which the look-up tables were pre-built for a set of assumptions. Every hydrometeor specie present in the cloud model output must be assigned to the corresponding scattering type in the CR-SIM configuration setting. This approach enables addition of the new "scattering types" obtained using different and more complex scattering assumptions (e.g., Kneifel et al., 2017; Leinonen and Moisseev, 2015; Leinonen and Szyrmer, 2015; Lu et al., 2016) without any change to the code.

We added these descriptions in summary and section 2. Thank you for the suggestions and pointing out those papers.

**Line 90: and spectrum width?**

Yes. We added spectrum width to the sentence.

**Line 105: What particle size spacing is used and what are the minimal and maximum sizes of particles simulated? The truncation can affect the resulting simulated measurements.**

We set the minimum and maximum sizes and size spacing of simulated particles for each hydrometeor category within the bulk microphysics schemes, except for the P3 scheme. 'Particle size' here refers to the particle maximum dimension. We added the information to the revised manuscript.

| Category | Minimum size [$\mu m$] | Maximum size [$\mu m$] | Size spacing [$\mu m$] |
|---|---|---|---|
| Cloud | 1 | 250 | 1 |
| Drizzle | 1 | 250 | 1 |
| Rain | 100 | 9000 | 20 |
| Ice | 1 | 1496 | 5 |
| Snow, aggregates | 100 | 50000 | 100 |
| Graupel | 5 | 50005 | 100 |
| Hail | 5 | 50005 | 100 |

**Lines 105 – 114: In general, I could follow the authors' description of the scattering properties and implement it into a simulator. However, there is no description of how mixed phase hydrometeors are treated. How is this accomplished?**

Please see our response to the referee's comment 1c.

**Line 197: Suggest "for convective cells" since multiple convective cells are evident in the image**

Done.

**Line 251: Should add a reference for the Morrison microphysics scheme**

Done. We added Morrison et al. (2005).

**Line 282: It isn't clear which simulation output is saved every 10 minutes (CR-SIM or WRF LES), or both.**

The WRF LES output is saved every 10 minutes, and CR-SIM is run for each output file. We revised the sentence.

**Lines 289 – 290: Is spatial or temporal sampling driving these major errors?**

We think that both can be major sources of the errors. We revised the sentence to read "the limited spatial and/or temporal sampling is the major error source to consider when comparing the profiling measurement derived CFP with the domain-averaged WRF CFP."

**Line 332: Should this say minimum detectable reflectivity Zmin, similar to a KASCR? Can you please provide the sensitivity of the simulated radar?**

The minimum detectable reflectivity $Z_{MIN}$ in the simulation is given by Eq. (4) with a constant C=-50 dBZ, which is similar to the Ka-band ARM scanning cloud radar (KaSACR). We revised the sentence.

**Lines 342 – 343: How does this compare to the current CWRHI measurement interval?**

The CFP estimation technique was applied to the Ka-band ARM scanning cloud radar at SGP. The product is available from the ARM Archive as an evaluation stage (https://iop.archive.arm.gov/arm-iop-file/0eval-data/wang/kasacradv3d3c/README.html). The CWRHI scans used in the product were provided at 20 sec intervals, which is a higher temporal resolution than that assumed in this study.

**Lines 351 – 352: References needed for multi-Doppler error sources**

We referred to Clark et al. (1980), Bousquet et al. (2008), and Potvin et al. (2012) in the revised manuscript.

**Line 356: Which advection-correction technique? Please state and cite**

Oue et al. (2019) used the advection-correction technique proposed by Shapiro et al. (2010) that allows for trajectories of multiple individual clouds, performs smooth grid-box-by-grid-box corrections of cloud locations, and takes into account changes in cloud shape with time by using PPI scans at two times. We added this description and reference to the revised manuscript.

**Lines 364 – 366: While this study is examining a hypothetical scenario for VCPs, is the 60 elevation angle scenario practical for the listed update intervals of 2 and 5 minutes? This would require PPI scans every 2 or 5 seconds which is not possible with the XSAPR (but is with other X-band radar systems). Please elaborate on the technology limitation.**

The referee is right. A volume scan with 60 elevations in less than 5 minutes is challenging for conventional scanning radars. The improvements required in the sampling strategy of the observing system (higher maximum elevation angle, higher density elevation angles and rapid VCP time period) can be accomplished using rapid scan radar systems such as the Doppler on

Wheels mobile radars (DOWs, Wurman, 2001) or even phased array radar systems (e.g. Kollias et al, 2018). We added this description to the last paragraph in section 3.4 in the revised manuscript.

**Figure 1: The Doppler velocity and spectrum with colors are saturated in a large portion of the figures. Suggest a wider colorbar range.**

Done.

**Technical Corrections: Line 280: Suggest "highly heterogeneous" instead of "high heterogenous"**

Done.

**Line 283: Suggest "between 10-minute intervals"**

Done.

**Line 302: Suggest "Each panel shows that CFPs at a single site"**

Done.

**Line 695: Extra comma in the data "May, 20, 2011"**

Done. We removed a comma after "May."

**Figure 7 caption: units for cloud water mixing ratio should be g/kg**

Done. Thank you for pointing the typo out.

**Figure 9: Suggest "20-second output for 5 minutes" to be more clear in the Forward Model box**

Done.

---

## Short Comment (SC1) · 25 Nov 2019

This is an executive editor comment highlighting the ways in which this manuscript is not currently compliant with GMD policy on code and data availability. In this case, there are a number of technical issues which needs to be remedied in the revised submission:

1. Model code on institutional websites. This is insufficiently persistent as institutional websites change. Please upload the exact version of the source code used to a persistent public archive such as Zenodo or the Stonybrook academic com-

mons, and cite it appropriately. Since the code is GPL, there should be nothing preventing this from being done.

2. Code available on request. I recognise that this is quite a small piece of code, but it breaks the provenance chain for the paper. Please archive this code somewhere suitable (if it is really rather small then you might just include it in the supplementary material of the paper).

3. The LASSO data used is not identified with sufficient precision that someone could reuse your work. The ARM archive provides a mechanism to generate a DOI for the exact data you want to cite. Please use this facility and cite the data following the instructions at https://www.arm.gov/working-with-arm/acknowledging-arm/doi-guidance-for-datastreams.

4. The archive of configuration files is excellent, and the Stonybrook academic commons complies with GMD policy. However, citing this by URL is not good practice. If you look at the entry in the repository itself, it shows you how to cite it. Please do so: https://commons.library.stonybrook.edu/somasdata/3/

Further details on code and data availability requirements are in the GMD model code and data policy: https://www.geoscientific-model-development.net/about/code_and_data_policy.html. The reasons for the policy and more detail are provided in this editorial: https://doi.org/10.5194/gmd-12-2215-2019.

---

## Referee Comment (RC2) · Anonymous Referee #2 · 28 Nov 2019

The manuscript introduces a software (CR-SIM) for simulating ground-based radar and lidar observations, based on input from atmospheric models. The software itself is presented and several possible applications are demonstrated. Tools of this type are needed to e.g. plan measurement campaigns and evaluate models using real observations. Accordingly, there exist important objectives and the manuscript fits GMD well.

As far as I can judge (with no direct experience of data of the type targeted by the software), the application examples are described sufficiently well. At least, the number of "use cases" is sufficiently high to convince a reader about the value of the software. On the other hand, I find the description of the features and limitations of the software

too short. I fully understand that not all details can be considered (but are hopefully covered by the user guide), but basic facts should be clarified in the manuscript, acting as the entrance points for potential users.

First of all, it should more clearly be expressed how CR-SIM relates to similar software. Is there any other software that can do the same things as CR-SIM? Is CR-SIM unique in any way? Further, the use of "Finally" on line 84 gives the impression that the review of other software is complete, but I strongly doubt that is the case. For example,

Matsui, T., Dolan, B., Rutledge, S. A., Tao, W.‐K., Iguchi, T., Barnum, J., & Lang, S. E. (2019). POLARRIS: A POLArimetric Radar Retrieval and Instrument Simulator. Journal of Geophysical Research: Atmospheres, 124, 4634–4657. https://doi.org/10.1029/2018JD028317

seems to have a similar scope as CR-SIM but is not mentioned.

The output variables should be better defined. For the radar ones (Table 2) not even the units are given. The dielectric factor used in the conversion to reflectivity can be defined in different ways. Does CR-SIM allow different options, or what option is used? Equations or citations for the relationship between the scattering matrix elements and the output variables should be given (see e.g. Eqs. 1-16 in Matsui et al.).

It is said that propagation effects are not treated. What is included in the term "propagation effects"?

Are there any other limitations that should be mentioned? As far as I understand, attenuation due to gases is not considered. That should be a significant effect at 94 GHz. Would be good to clarify if the attenuation due to liquid cloud droplets is included in the attenuation terms. Is the surface assumed to be flat or curved? Is refraction of importance? Ice particles seem to be treated as spheroids consisting of a mixture of ice and air. Just the choice of mixing rule (that is not specified) causes modelling uncertainties.

As a user, you need an estimate on the overall modelling uncertainty. For example, are differences between real observations and simulations of 3 dBZ significant or not?

I found the manuscript hard to read due to the high usage of acronyms. Consider if some acronyms can be avoided, or adding a table of acronyms. Specific comments:

Line 88: What do you mean with "quality-controlled" and how do you ensure it?

Line 94: T-matrix and DDA are general methods to calculate scattering properties, not scattering datasets. Is there any scattering dataset that could be coupled to your model?

Line 108: How is bulk density defined?

Line 138: Do you get the fall speed from the models, or by an external expression? If the later, add a reference.

Lines 153-154: I don't get what you want to say what this sentence.

Line 196 and elsewhere: I don't think you can expect that all readers know the frequency of the radar bands (C, X, ...). At least define at the first usage of each band.

Line 223: Start a new paragraph at "Figure 5 ..."

Line 238: "affects" -> "effects".

Lines 294-297: I could not understand this description.

Line 331: Is CWRHI something built into CR-SIM, or done by external processing?

Line 441: Is not the basic output from scanning radars in polar coordinates? If yes, is not this code essential to use CR-SIM and should then be fully integrated, as you claim that CR-SIM output "can be easily compared with real observations"?

---

## Author Comment (AC2) · 4 Dec 2019

**Response to comments from Executive Editor**

**This is an executive editor comment highlighting the ways in which this manuscript is not currently compliant with GMD policy on code and data availability. In this case, there are a number of technical issues which needs to be remedied in the revised submission:**
**1. Model code on institutional websites. This is insufficiently persistent as institutional websites change. Please upload the exact version of the source code used to a persistent public archive such as Zenodo or the Stonybrook academic commons, and cite it appropriately. Since the code is GPL, there should be nothing preventing this from being done.**

We have submitted the source code to the Stony Brook Academic Commons. However, we have not received the right link of the code from the university yet. We will update the link as soon as we receive it. Because the source code is the latest version 3.3, we changed the title to "The Cloud Resolving Model Radar Simulator (CR-SIM) Version 3.3: Description and Applications of a Virtual Observatory."

**2. Code available on request. I recognise that this is quite a small piece of code, but it breaks the provenance chain for the paper. Please archive this code somewhere suitable (if it is really rather small then you might just include it in the supplementary material of the paper).**

We posted the code that converts model grid coordinate to radar polar coordinate at the CR-SIM website on August 30, 2019. In addition, the code has been submitted to the Stony Brook Academic Commons together with the CR-SIM package. We have updated the code availability.

**3. The LASSO data used is not identified with sufficient precision that someone could reuse your work. The ARM archive provides a mechanism to generate a DOI for the exact data you want to cite. Please use this facility and cite the data following the instructions at https://www.arm.gov/working-with-arm/ acknowledging-arm/doi-guidance-for-datastreams.**

We have cited:
Atmospheric Radiation Measurement (ARM) Research Facility. September 2017. LASSO Data Bundles. , 36° 36′ 18.0″ N, 97° 29′ 6.0″ W: Southern Great Plains Central Facility (C1). Compiled by WI Gustafson, AM Vogelmann, X Cheng, S Endo, KL Johnson, B Krishna, Z Li, T Toto, and H Xiao. ARM Data Archive: Oak Ridge, Tennessee, USA. Data set accessed at http://dx.doi.org/10.5439/1342961.

**4. The archive of configuration files is excellent, and the Stonybrook academic commons complies with GMD policy. However, citing this by URL is not good practice. If you look at the entry in the repository itself, it shows you how to cite it. Please do so: https://commons.library.stonybrook.edu/somasdata/3/ Further details on code and data availability requirements are in the GMD model code and data policy: https://www.geoscientific-model-development.net/about/code_ and_data_policy.html. The**

**reasons for the policy and more detail are provided in this editorial: https://doi.org/10.5194/gmd-12-2215-2019.**

Thank you pointing this out. The data submission was posted on Oct. 4, 2019. We have included the right URL and citation in the revised manuscript.

---

## Author Comment (AC3) · 14 Dec 2019

We would like to update the data and code availability.

Executive Editor's comment: 1. Model code on institutional websites. This is insufficiently persistent as institutional websites change. Please upload the exact version of the source code used to a persistent public archive such as Zenodo or the Stonybrook academic commons, and cite it appropriately. Since the code is GPL, there should be nothing preventing this from being done.

Response updated:    The  source  code  has  been  posted  to  the  Stony  Brook

[Figure]

Academic Commons on December 9, 2019. The link of the code is https://commons.library.stonybrook.edu/somasdata/4. We have updated the code availability in the revised manuscript.

Executive Editor's comment: 2. Code available on request. I recognise that this is quite a small piece of code, but it breaks the provenance chain for the paper. Please archive this code somewhere suitable (if it is really rather small then you might just include it in the supplementary material of the paper).

Response updated: The code that converts model grid coordinate to radar polar co-ordinate has been posted to the Stony Brook Academic Commons on December 9, 2019. The data link is the same as the CR-SIM source code. We have updated the code availability in the revised manuscript.

---

## Author Comment (AC4) · 14 Dec 2019

**The manuscript introduces a software (CR-SIM) for simulating ground-based radar and lidar observations, based on input from atmospheric models. The software itself is presented and several possible applications are demonstrated. Tools of this type are needed to e.g. plan measurement campaigns and evaluate models using real observations. Accordingly, there exist important objectives and the manuscript fits GMD well. As far as I can judge (with no direct experience of data of the type targeted by the software), the application examples are described sufficiently well. At least, the number of "use cases" is sufficiently high to convince a reader about the value of the software. On the other hand, I find the description of the features and limitations of the software too short. I fully understand that not all details can be considered (but are hopefully covered by the user guide), but basic facts should be clarified in the manuscript, acting as the entrance points for potential users.**

We thank the referee for their time and consideration reviewing the manuscript. We have revised the manuscript addressing all comments. Please see our point-by-point responses to the referee's comments.

**1. First of all, it should more clearly be expressed how CR-SIM relates to similar software. Is there any other software that can do the same things as CR-SIM? Is CR-SIM unique in any way? Further, the use of "Finally" on line 84 gives the impression that the review of other software is complete, but I strongly doubt that is the case. For example, Matsui, T., Dolan, B., Rutledge, S. A., Tao, W.- K., Iguchi, T., Barnum, J., & Lang, S. E. (2019). POLARRIS: A POLArimetric Radar Retrieval and Instrument Simulator. Journal of Geophysical Research: Atmospheres, 124, 4634–4657. https://doi.org/10.1029/2018JD028317 seems to have a similar scope as CR-SIM but is not mentioned.**

Thank you for informing us about the paper to update the abstract citation originally used. In the revised manuscript, we refer to POLARRIS and this paper in Section 2 instead of Dolan et al. (2017) and added a sentence "Matsui et al. (2019) simulated polarimetric precipitation radar-based hydrometeor classification, vertical velocity, and rain rate from CRM output to examine uncertainties in the retrieval algorithms and model microphysical parameterizations using POLARRIS."

We also compare CR-SIM with other simulators in section 4. Please see our response to referee #1's specific comment #1.

We agree with the referee that there are many other radar simulators and thus the list of existing software in this manuscript cannot be all-inclusive. We removed the word "Finally" and added "For example" to the beginning of the third sentence in Section 2.

**2. The output variables should be better defined. For the radar ones (Table 2) not even the units are given. The dielectric factor used in the conversion to reflectivity can be defined in different ways. Does CR-SIM allow different options, or what option is used? Equations or citations for the relationship between the scattering matrix elements and the output variables should be given (see e.g. Eqs. 1-16 in Matsui et al.).**
**It is said that propagation effects are not treated. What is included in the term "propagation effects"?**

The radar observables are computed by integrating scattering properties over the discrete PSD using a constant bin size for each hydrometeor. We followed the microphysical parameterization of the selected microphysics scheme to retrieve the PSD. The complex scattering amplitudes of the 2 x 2 scattering matrix are pre-computed for equally spaced particle sizes and stored in the look-up tables using the Mishchenko's T-matrix code for single non-spherical particles. Using the calculated scattering amplitudes stored in the LUTs, we computed radar observables following Ryzhkov et al. (2011), using for an assumption of the orientation distribution. CR-SIM incorporates three options of the orientation distribution model which can be selected by the user. The composition of particles is accounted for in the scattering computations by an appropriate selection of the dielectric constant for different hydrometeor types. The dielectric constant of liquid particles is frequency- and temperature-dependent (Ray, 1972). Ice phase hydrometeors are assumed to be made of ice inclusions in an ice matrix and their effective dielectric constant is computed using the Maxwell-Garnet mixing formula (1904). The output $Z_{hh}$ is the equivalent radar reflectivity, in which computations, we adopt 0.92 as the value of dielectric factor for liquid water at centimeter wavelengths. This choice of the dielectric factor is dictated by convention to ensure that the definition of radar reflectivity reduces to form: $Z = \int N(D)D^6 dD$ for (spherical) liquid particles, where D is the droplet diameter and N(D) is the droplet size distribution function.

The above information has been added to section 2 in the revised manuscript. Because the equations of radar observables have been well described in Ryzhkov et al. (2011), we decided to not add them in the manuscript. The method is fully described in the Section 4.4 and 4.5 in the CR-SIM User Guide (ftp://ftp.radar.bnl.gov/outgoing/moue/crsim/docs/crsim-UserGuide-v3.3.0.pdf).

We also added units for each output radar variable in Table 2 (now Table 3).

We meant "propagation effects" as the total attenuation along a radar beam path. We include specific attenuation from all simulated hydrometeors (i.e., cloud droplets, cloud ice, rain, snow, and graupel for the analysis in the manuscript), but gaseous attenuation was not included. We added the information in section 2.3.

**3. Are there any other limitations that should be mentioned? As far as I understand, attenuation due to gases is not considered. That should be a significant effect at 94 GHz. Would be good to clarify if the attenuation due to liquid cloud droplets is included in the attenuation terms. Is the surface assumed to be flat or curved? Is refraction of importance? Ice particles seem to be treated as spheroids consisting of a mixture of ice and air. Just the choice of mixing rule (that is not specified) causes modelling uncertainties.**

CR-SIM treats hydrometer categories for which mixing ratio (and/or number density) are provided by the input cloud model using the selected microphysics scheme. At this stage, all ice hydrometeors (e.g., snow, ice, graupel, hail) are modeled as dielectrically dry spheroids i.e., assuming the dry growth of larger ice particles. Thus, the refractive index of ice-phase hydrometeors depends on relative mixture of air and solid ice and is computed using the Maxwell-Garnet mixing formula (1904). The LUTs of scattering properties incorporated in the current CR-SIM were created using the T-matrix method for selected assumptions regarding ice particle composition and shape. More complex electromagnetic scatters can be incorporated by

adding LUTs of their scattering properties from different scattering calculation methods without any change to the code. We stated this in the revised manuscript in Section 2.1.

As in the response to the previous comment #2, we did not calculate gaseous attenuation in CR-SIM. We thank you for this suggestion. We will certainly add water vapor attenuation in a future version of CR-SIM.

In CR-SIM, the earth surface is assumed to be flat. This assumption is acceptable for shorter observation ranges. The description is added in section 2.3 in the revised manuscript. However, this could be a source of uncertainty for longer distances from the radar and/or for small model vertical grid spacings. We investigated the differences between the two assumptions (earth curvature + atmospheric refraction, and a flat surface). The figure below shows elevation angles at a height of 5.5 km AGL as a function of horizontal distance (right) and height at an elevation angle of 2º as a function of radar range (left). Here, the Earth's surface is represented using a sphere with a radius of 6370.0 km for red lines or flat for blue lines. The black lines represent the difference between the two assumptions. The difference in elevation angle is less than 1 degree for the horizontal distance less than 300 km at a fixed height. This is smaller than the elevation spacing of the scattering LUTs. At a fixed elevation angle, the difference in height is less than 1.4 km for the radar ranges < 150 km and is greater than 5 km at the radar range of 300 km. If the model vertical spacing is smaller than the difference, the error from the flat surface assumption could be significant. In the MCS simulation presented in the manuscript, the maximum radar range is 50 km for the X-band radars, and the model vertical resolution is approximately 250 m. For the shallow cumulus cases in the manuscript, the maximum radar range is 15 km and the model vertical resolution is 30 m. The differences at the maximum radar range for those cases are smaller enough than the model vertical spacings. However, we need to carefully configure the simulation settings considering the uncertainty. We thank the referee for their comment.

[Figure]

Figure 1: (Left) Elevation angle at a fixed height of 5.5 km AGL as a function of horizontal distance from the curved surface including the atmospheric refraction (blue) and flat surface (red) assumptions. (Right) Height at a fixed elevation angle of 2º as a function of radar range. Black lines represent the difference between the two assumptions. The curved surface is assumed to be a sphere surface with a radius of 6370.0 km.

**4. As a user, you need an estimate on the overall modelling uncertainty. For example, are differences between real observations and simulations of 3 dBZ significant or not?**

There are several sources of uncertainty in CR-SIM including particle composition and shapes, particle size, and canting angles for the calculations of the single scattering properties. We have

investigated the uncertainties in some of these assumptions. The figure below shows statistical differences of simulated radar observables between two particle shape assumptions: spheroid ice (with an aspect ratio of 0.6) and spherical ice, using WRF+P3 model outputs. As expected, the aspect ratio assumption has the most influence on the polarimetric observables (i.e., $Z_{DR}$ and $K_{DP}$). The influence is larger for the unrimed ice particles than for the partly-rimed ice particles.

As the referee pointed, the assumption of a flat Earth can also be an uncertainty source. Please see our response to the previous comment.

In addition, because CR-SIM ingests numerical model output, the assumptions in the microphysics schemes could also contribute to the uncertainty in CR-SIM. For example, the truncation of the edges of the PSDs (e.g., minimum and maximum sizes, size spacing) could affect CR-SIM results as pointed by referee#1. We added the settings of the PSD in Table 1 in the revised manuscript.

Since the input numerical model simulation itself can also include uncertainties, it is difficult to find an intrinsic uncertainty in CR-SIM by comparing the CR-SIM results with real observations. But comparisons with different radar simulator's results can help to understand this uncertainty. One CR-SIM user compared the CR-SIM simulated reflectivity with reflectivity from another simulator (Passive and Active Microwave TRANsfer, PANTRA) for the same model input data. He reported differences in reflectivity ranging from 0.5 to 2 dB probably due to a difference of the diameter spacing implemented in the simulators.

We plan to summarize the uncertainty analyses in a follow-up paper.

[Figure]

Figure 2. Box and whisker plots of the changes in (a) $Z_{HH}$, (b) $Z_{DR}$, and (c) $K_{DP}$ by assuming spheroidal ice particles (aspect ratio = 0.6), compared to spherical particles (aspect ratio = 1.0) with a fixed canting angle of 0 degree at a radar frequency of 3 GHz. The results are shown for ice categories (partly rimed ice, unrimed ice, and total ice) predicted by WRF P3 microphysics scheme for the mesoscale convective system simulation. Lower and upper box boundaries are 25th and 75th percentiles, respectively, the lines inside the box are medians, and the outermost lower and upper lines are 10th and 90th percentiles, respectively. The radar elevation angle is 20⁰ for all simulations.

**I found the manuscript hard to read due to the high usage of acronyms. Consider if some acronyms can be avoided, or adding a table of acronyms.**

Thank you for this suggestion. We added a section with a list of acronyms.

**Specific comments:**
**Line 88: What do you mean with "quality-controlled" and how do you ensure it?**
Here, we mean ideal values of radar observables without observational limitations such as sensitivity, minimum detectable value, or hydrometeor attenuation. We realized that "quality-controlled" was a confusing word and used "ideal" instead in the revised sentence.

**Line 94: T-matrix and DDA are general methods to calculate scattering properties, not scattering datasets. Is there any scattering dataset that could be coupled to your model?**

We modified the phrase to read "scattering methods." We added a description of the possibility to couple with different scattering calculation methods in section 2.1. Different scattering calculation datasets can be easily incorporated by adding LUTs of their calculated particle scattering properties (e.g., Kneifel et al., 2017; Leinonen and Moisseev, 2015; Leinonen and Szyrmer, 2015; Lu et al., 2016) without any change to the code. Please also see our response to the referee #1's specific comment #1.

**Line 108: How is bulk density defined?**

The bulk density used is as parameterized in the selected microphysics scheme, assuming spheroidal particle shapes. Change made.

**Line 138: Do you get the fall speed from the models, or by an external expression? If the later, add a reference.**

We calculate the fall speed following the same manner as in the selected microphysics scheme in the models (given on L159). The parameterization depends on the microphysics schemes that are described in the references in (the now) Table 2 in the revised manuscript.

**Lines 153-154: I don't get what you want to say what this sentence.**

We revised the sentence to read "As expected, the lidar backscatter is significantly attenuated by cloud droplets, but the very high lidar backscatter at the interface between air a cloud can be used to detect cloud base height."

**Line 196 and elsewhere: I don't think you can expect that all readers know the frequency of the radar bands (C, X, ...). At least define at the first usage of each band.**

Thank you for the suggestion. We added the radar bands in text and figure captions.

**Line 223: Start a new paragraph at "Figure 5 ..."**

Done.

**Line 238: "affects" -> "effects".**

Done.

**Lines 294-297: I could not understand this description.**

We improved the paragraph. First, observation sites are randomly selected within the horizontal domain. Second, for each snapshot of the simulation, clouds over the observation sites are sampled as if the clouds are frozen in time and advected by the mean environmental wind. Thus, the columns are sampled along the direction of the horizontal wind over the advected distance (i.e. horizontal wind speed x 10 min), where the environmental horizontal wind at each snapshot is the mean horizontal wind across the simulation domain within the cumulus cloud layer (i.e., the layer between the mean cumulus cloud base and the maximum cumulus cloud top). Third and last, the CFP is estimated by varying both the number of observation sites and the integration period.

**Line 331: Is CWRHI something built into CR-SIM, or done by external processing?**

To simulate CWRHI accounting for radar beamwidth and range-gate spacing, a post-processing code is required. However, in the analysis by Oue et al. (2016), a CWRHI scan was referred to as a vertical cross section of CR-SIM simulated radar observables at the model grid. Figure 8 accounts for $Z_{MIN}$ only. Details are described in Oue et al. (2016). To avoid the confusion, we revised the sentences in this paragraph.

**Line 441: Is not the basic output from scanning radars in polar coordinates? If yes, is not this code essential to use CR-SIM and should then be fully integrated, as you claim that CR-SIM output "can be easily compared with real observations"?**

The CR-SIM standard output is radar and lidar observables for all the cloud resolving model grid boxes accounting for elevation angles relative to a radar location, not polar coordinates that account for radar geometric characteristics such as beamwidth and radar range resolution to simulate scatters within the radar resolution volume. We describe this in Section 4 in the revised manuscript. The post-processing instrument model accounts for the radar sampling characteristics and outputs the observables in the polar coordinates. The post-processing code is now publicly available at the CR-SIM website (https://you.stonybrook.edu/radar/research/radarsimulators/) and also submitted to the Stony Brook University Academic Commons.

This feature facilitates the process of configuring any desirable observational setup with a varying number of profiling or scanning sensors and makes the gridbox-by-gridbox comparisons of the ideal radar variables easy.

---

## Author Response (AR2)

Response to Topical Editor

Dear Dr. Simon Unterstrasser,

Thank you very much for your careful review of our manuscript. We have modified the manuscript, taking into account your suggestions and the comments from Reviewer #2. The following text contains our detailed responses to Topical Editor and Reviewer comments, with our responses in plain type given underneath the original comments in bold.

**I share the opinion of the reviewer, that more substantial changes to the manuscript should have been incorporated.**

We have considered the Reviewer's input and revised the manuscript.

**In addition to the present reviewer remarks, I think a paragraph on the computing time would be of interest to the audience (you answered the question only in your reply).**

We added the following sentences to Section 2.4.
"The runtime depends on the computing power, number of threads used, simulation domain size, and the number of cloudy grid boxes. The simulations presented in this manuscript were run on a computer having 500 GB memory and 24 processors each with 12 cores. For the MCS case in Fig. 1, having a simulation domain size of $667 \times 667 \times 12$ (5.3M grid points), the runtime is approximately 270 seconds using 16 threads. "

**Moreover, I noticed that the language, in particular, in the added parts could be improved. I am sure there are English native speakers in the author team who can take care of it.**

We have carefully checked the language in the text.

Response to Reviewer #2

**Going back to my initial review, I want to clarify that with "basic facts should be clarified in the manuscript" I did not just mean adding details in the text, but that this should be considered throughout. Most importantly, I find both abstract and summary non-informative. They should contain more facts. Especially I would like these parts to clarify the novelty of the software, but also the main limitations should be clearly stated (at least in summary). That is, I should get a fairly good feeling by only reading abstract and summary if this tool is of interest for me, if I plan some simulations in this direction. Or if I shall look for another software?**

We revised the abstract and summary to highlight CR-SIM's advantages and limitations. In particular, we added the following text to the summary:
"CR-SIM is easily expanded to include additional microphysical schemes, new scattering classes, scattering calculations, and other applications to simulate multi-sensor products (e.g., multi-Doppler wind retrievals, ARSCL). At this stage, all ice hydrometeors (e.g., snow, ice, graupel, hail) are modeled as dielectrically dry spheroids. The LUTs of scattering properties incorporated in the current CR-SIM were created using the T-matrix method based on selected assumptions regarding ice particle composition and shape. More single-scattering properties from other scattering calculation methods can be incorporated by adding LUTs. Moreover, the gaseous attenuation will be considered in the future, as the gaseous attenuation effect can be significant in the millimeter-wavelength radar measurements, and elevation angles will be corrected for the Earth's curvature."

We also revised the abstract to add "CR-SIM allows direct comparison between an atmospheric model simulation and remote-sensing products using a forward-modeling framework consistent with the microphysical assumptions used in the atmospheric model. CR-SIM has the flexibility to easily incorporate additional microphysical modules, such as microphysical schemes and scattering calculations, and expand the applications to simulate multi-sensor retrieval products."

**I wrote "I fully understand that not all details can be considered", but I tried to also indicate that it would still be good to know if the information is available at all. That is, I would recommend to explain if the user guide provides further information for the different parts. I have not looked into the user guide, I don't think that should be required to review the manuscript.**

Thank you. We believe that the user guide does provide the desired further information. Additionally, to make the manuscript more self-contained, we added equations of radar backscatter calculations used in CR-SIM to the Appendix in the revised manuscript, which were previously referenced to the user guide.

**Examples on parts that need further description, either by extending the manuscript or making clear that the information is found in the user guide:**

**- Calculation of particle fall velocity. For example, how has the parameterisation been determined? Does the fall velocity depend on assumed particle orientation?**

The following description has been added to Section 2.2.

"The fall velocity size relationship ($V_f(D)$) for each hydrometeor specie is specified in a form:
$$V_f(D) = f_c a_v D^{b_v} \tag{1}$$
where $f_c = \left(\rho_{surf}/\rho\right)^k$ is the correction factor for air density with exponent $k$. The values for the prefactor $a_v$, the exponent $b_v$, and the exponent $k$ vary according to the microphysics scheme and do not depend on particle orientation. The air density at sea level, $\rho_{surf}$, is computed for the first model level. The values of coefficients and detailed descriptions concerning the microphysics schemes are found in Tatarevic et al. (2019)."

**- Information about dielectric constant for liquid particles has been added, but the corresponding information for ice is missing.**

The effective dielectric constant is computed using the Maxwell-Garnet mixing formula (Maxwell Garnet, 1904). To compute the equivalent radar reflectivity moments, we adopted the dielectric factor of 0.92 for all hydrometeors. We described this in Section 2.1.

**- The part on ceilometer and lidar simulations is extremely brief.**

We added equations of how the lidar backscattering is calculated in CR-SIM to the Appendix in the revised manuscript.

**Example on parts that have been modified but are still hard to follow:**
**\* Please explain clearly what you mean with terms such as propagation-corrected and propagation effects, the first time they are used.**

Propagation effect in the manuscript means total (two-way) attenuation. We reworded this phrase "total (two-way) attenuation" in the revised manuscript. "Propagation-corrected" is no longer used.

**\* Text around line 95 and 146 indicates that external scattering properties can be included, while elsewhere a very rigid format for the scattering properties is described that seems to contradict the use of external data. On line 101 it even says that the LUTs are calculated by the T-matrix method. Even if this is poorly expressed and external data can be imported, it is highly unlikely that the external data match the orientations, size ranges and bin spacings that seem to be hard-coded. Is there any flexibility to facilitate to use external data?**

The scattering properties in the LUTs were pre-built for single particles by setting assumptions of particle states such as density, phase, temperature, shape (aspect ratio), and size. These assumptions depend on the "scattering type," which must be assigned to the corresponding hydrometeor category present in the model simulation output. The scattering types and their assumptions are presented in Table 2 in the revised manuscript. The LUTs can be expanded by adding new scattering types, where additional assumptions (e.g., phase, shape) can be accounted for in the scattering calculations. New modules of orientation can be incorporated if the input data uses a different orientation assumption. The only rigid requirement is that the elements of the scattering matrix must be computed for equally spaced particle sizes in the predefined diameter range.

[revised manuscript text omitted]